# Composition and origin of PM$_{2.5}$ aerosol particles in the upper Rhine valley in summer

Xiaoli Shen[1,2], Heike Vogel[1], Bernhard Vogel[1], Wei Huang[1,2], Claudia Mohr[3], Ramakrishna Ramisetty[1,4], Thomas Leisner[1,5], André S. H. Prévôt[6], and Harald Saathoff[1,*]

[1]Institute of Meteorology and Climate Research, Karlsruhe Institute of Technology, Hermann-von-Helmholtz-Platz 1, 76344 Eggenstein-Leopoldshafen, Germany
[2]Institute of Geography and Geoecology, Working Group for Environmental Mineralogy and Environmental System Analysis, Karlsruhe Institute of Technology, Kaiserstr.12, 76131 Karlsruhe, Germany
[3]Department of Environmental Science and Analytical Chemistry, Stockholm University, Stockholm, 11418, Sweden
[4]Now at: TSI Instruments India Private Limited, Bangalore, 560102, India
[5]Institute of Environmental Physics, University Heidelberg, In Neuenheimer Feld 229, 69120 Heidelberg, Germany
[6]Laboratory of Atmospheric Chemistry, Paul Scherrer Institute (PSI), 5232 Villigen PSI, Switzerland

*Correspondence to*: Harald Saathoff (harald.saathoff@kit.edu)

**Abstract.** We conducted a six-week measurement campaign in summer 2016 at a rural site about 11 km north of the city of Karlsruhe in southwest Germany in order to study the chemical composition and origin of aerosols in the upper Rhine valley. In particular, we deployed a single particle mass spectrometer (LAAPTOF) and an aerosol mass spectrometer (AMS) to provide complementary chemical information of the aerosol particles smaller than 2.5 µm. For the entire measurement period, the total aerosol particle mass was dominated by sodium salts contributing on average $(36 \pm 27)$ % to the total single particles measured by the LAAPTOF. The total particulate organic compounds, sulfate, nitrate, and ammonium contributed on average $(58 \pm 12)$ %, $(22 \pm 7)$ %, $(10 \pm 1)$ %, and $(9 \pm 3)$ % to the total non-refractory particle mass measured by the AMS. Positive matrix factorization (PMF) analysis for the AMS data suggests that the total organic aerosol (OA) consisted of five components, including $(9 \pm 7)$ % hydrocarbon-like OA (HOA), $(16 \pm 11)$ % semi-volatile oxygenated OA (SV-OOA), and $(75 \pm 15)$ % low-volatility oxygenated OA (LV-OOA). The regional transport model COSMO-ART was applied for source apportionment and to achieve a better understanding of the impact of complex transport pattern on the field observations. Combining field observations and model simulations, we attributed high particle numbers and SO$_2$ concentrations observed at this rural site to industrial emissions from power plants and a refinery in Karlsruhe. In addition, two characteristic episodes with aerosol particle mass dominated by sodium salts particles compromising $(70 \pm 24)$ % of the total single particles and organic compounds comprising accounting for $(77 \pm 6)$ % of total non-refractory species, respectively, were investigated in detail. For the first episode, we identified relatively fresh and aged sea salt particles originating from the Atlantic Ocean more than 800 km away. These particles showed markers like m/z 129 C$_5$H$_7$NO$_3^+$ indicating the influence of anthropogenic emissions modifying their composition e.g., from chloride to nitrate salts during the long-range transport. For a 3-day episode including high organic mass concentrations, model simulations show that on average $(74 \pm 7)$ % of the particulate organics at this site were of biogenic origin. Detailed model analysis allowed us to find out that three subsequent peaks of high organic mass concentrations originated

from different sources, including local emissions from the city and industrial area of Karlsruhe, regional transport from the city of Stuttgart (~64 km away), and potential local night-time formation and growths. Biogenic (forest) and anthropogenic (urban) emissions were mixed during transport and contributed to the formation of organic particles. In addition, topography, temperature inversion, and stagnant meteorological conditions also played a role in the build-up of higher organic particle mass concentrations. Furthermore, the model was evaluated using the field observations and corresponding sensitivity tests. The model results show good agreement with trends and concentrations observed for several trace gases (e.g., $O_3$, $NO_2$, and $SO_2$) and aerosol particle compounds (e.g., ammonium and nitrate). However, the model underestimates the number of particles by an order of magnitude and underestimates the mass of organic particles by a factor of 2.3. The discrepancy was expected for particle number since the model does not include all nucleation processes. The missing organic mass indicates either an underestimated regional background, missing sources, and/or mechanisms in the model like night-time chemistry. This study demonstrates the potential of combining comprehensive field observations with dedicated transport modelling to understand the chemical composition and complex origin of aerosols.

## 1 Introduction

Aerosols are ubiquitous in the atmosphere and play a crucial role in air quality, human health, ecosystem and climate change. In particular, aerosol particles modify atmospheric radiative fluxes and can serve as cloud condensation nuclei (CCN) and/or be activated as ice nuclei (IN), which influences the cloud formation, properties (albedo) and precipitation, thus has further impact on weather and climate change (Fuzzi et al., 2015; Pöschl, 2005; Seinfeld and Pandis, 2006). Such indirect impacts of aerosols on climate are still related to large uncertainties and low confidence levels (IPCC, 2013). There are various types of atmospheric aerosol particles, such as mineral dust, sea salt, soot, and organic aerosol particles, originating from biogenic and/or anthropogenic emission sources, as well as secondary formation from precursors within the atmosphere (Tsigaridis et al., 2006). Aerosol particles are not typically purely inorganic or organic; instead, they are often composed of different inorganic and organic species. The diverse origins and atmospheric aging cause the complexity of the aerosol particle mixtures.

Real-time mass spectrometry is widely used to measure size and chemical composition of aerosol particles with high temporal resolution. Two major instrument types are single particle mass spectrometers (singe-particle MS) and aerosol mass spectrometers (AMS). The single-particle MS is sensitive to both non-refractory and refractory species and to the mixing state of individual particles, and thus identifies particle types (Murphy, 2007), while the AMS can quantify mass concentrations of non-refractory particulate compounds in bulk samples (Canagaratna et al., 2007). Complementary data from single-particle MS and AMS allows quantitative interpretation of the internal and external mixing state of the aerosol (Gemayel et al., 2017; Gunsch et al., 2018; Healy et al., 2013; Jeong et al., 2011; Shen et al., 2019; Zhou et al., 2016). The mixing state of the aerosol particles can reflect their evolution e.g. caused by chemical reactions leading to changing compositions or coatings (Reitz et al., 2016). A good example is the aging of sea salt particles: chloride replacement by nitrate on sea-salt particles can be well identified by single-particle MS (Gard et al., 1998).

The large diversity of aerosols, containing hundreds of gaseous and particulate components, makes it difficult to trace their sources. Therefore, several source apportionment methods have been developed with the common aim of reducing the complexity of the aerosol particle mixtures and thus helping to identify the major particle types in the atmosphere. Clustering algorithms have been used commonly for the single-particle MS data analysis (Reitz et al., 2016), while positive matrix factorization (PMF; Paatero, 1997; Paatero and Tapper, 1994) is commonly applied for AMS data analysis (Canonaco et al., 2013; Canonaco et al., 2015; Crippa et al., 2014; DeCarlo et al., 2010; Mohr et al., 2012; Ulbrich et al., 2009; Zhang et al., 2011). According to PMF analysis, the organic aerosol (OA) can be separated into e.g., hydrocarbon-like OA (HOA), cooking-related OA (COA), nitrogen-enriched OA (NOA), biomass burning OA (BBOA), and semi-volatile oxygenated OA (SV-OOA), and low-volatility oxygenated OA (LV-OOA). This information can give valuable hints to OA aging and potential sources. However, these methods can only provide indirect source information. The application of a transport model gives additional information. It can predict the state of the atmosphere based on detailed meteorology, emission data, as well as conversions of reactive species and thus allows studying the aerosol evolution from the emissions to a site of interest.

There exists a variety of so-called online coupled chemical transport models. A recent review is given by (Baklanov et al., 2014). Such models are important tools to better understand aerosol particles' sources and aging processes, and their impact on particles' climate-relevant properties, such as cloud condensation nuclei (CCN) activity, optical properties, and hygroscopicity (Fierce et al., 2017). Models can be improved with a good process understanding, e.g., emissions, transport and deposition, and corresponding parameterisations obtained from dedicated laboratory studies (Burkholder et al., 2017). They are often used to predict the evolution of the atmospheric composition on different spatial scales, which can be evaluated by comparison with field measurements. Besides, comparisons between model simulations and field observations also help to validate the model or to identify weaknesses like poorly represented chemistry, missing sources or mechanisms.

For comparison with measurements at a single location, regional scale models with small grid sizes (down to a few kilometres) are required. In addition, phenomena such as mountain and valley winds, land-sea breezes or lee waves become important, which requires a non-hydrostatic formulation of the model equations. In this study we use the regional-scale model system COSMO-ART (Consortium for small-scale modelling - aerosols and reactive trace gases; Vogel et al., 2009). The part of the model describing aerosols and reactive trace gases was developed at the Karlsruhe Institute of Technology (http://www.imk-tro.kit.edu/english/3509.php). It extends the operational weather forecast model COSMO of the German Weather Service (DWD), and it allows studying the feedback processes between aerosol particles and the state of the atmosphere covering the continental to the regional scale.

Mobile particle measurements on a tram line intersecting the city of Karlsruhe in southwest Germany and in the upper Rhine valley, showed particle number concentrations in the rural area north of the city almost as high as in downtown (particle size > 4 nm; mean value = $1.2 \times 10^4$ cm$^{-3}$, maximum values $3 \times 10^4$ cm$^{-3}$) (Hagemann et al., 2014). The Rhine valley provides a well-known wind channel of large-scale flow. This results in faster transport of pollutants along the valley axis, e.g., NO$_x$, SO$_2$, volatile organic compounds (VOCs) and particles (Gross and Wippermann, 1987; Hagemann et al., 2014; Kalthoff and Vogel, 1992; Vogel et al., 1986). According to data from the local air quality monitoring administration (LUBW,

https://www.lubw.baden-wuerttemberg.de/luft) from 2016 to 2018, the annual averaged mass concentrations of $NO_2$, $SO_2$, $PM_{10}$, and soot are 20, 2, 16, and 1.8 μg m$^{-3}$, respectively. They were measured at Eggenstein monitoring station (3.3 km south of the measurement site used in this study and about 9 km north of the city of Karlsruhe). Specifically, $NO_2$ concentrations are about half of the values measured in downtown Karlsruhe, but they exhibit comparable peak values (around 50–80 μg m$^{-3}$). $SO_2$ levels at Eggenstein are the second highest in the state of Baden-Württemberg (from the available data from 2016 to 2018), and

maximum hourly values can reach up to ~50 μg m$^{-3}$.

     In order to understand the sources for high local particle concentrations as well as the chemical nature and mass concentrations of aerosol types prevailing in the upper Rhine valley in summer-time we combine comprehensive aerosol information from point measurements with a regional transport model. Besides a better understanding of the origin of the aerosol in that region, we want to validate the capabilities of our transport model to predict characteristic air quality parameters.

**2 Methods**

**2.1 Measurement site and instrumentation**

The measurement site (49°6'10.54" N, 8°24'26.07" E) is in the upper Rhine valley 175 m north of the village of Leopoldshafen and 10 m east of the tramline No S1/11 described by Hagemann et al., (2014) (marked with a star in Fig. 1). The Rhine valley often acts as a wind channel leading to wind directions mainly from southwest or northeast. As shown in Fig. 1, the measurement

site is located ~11 km north of downtown Karlsruhe and ~5 to 15 km northeast of an industrial area, including a refinery with 15.5 Mt/yr capacity, a 1450 MW hard coal fired power plant, as well as a 365 MW gas fired power plant (Hagemann et al., 2014). Therefore, the measurement site is potentially subject to the urban and industrial emissions especially for southerly wind directions. Apart from that, there are several nearby biogenic emission sources, e.g., forests, sunflower fields, as well as other agricultural areas (Hagemann et al., 2014; Huang et al., 2019). Local traffic emissions from the minor road near the measurement

site was identified by measurements with high time resolution. Its contributions to the aerosol load are small due to the generally weak traffic.

     The instrumentation for this campaign has been described in detail in previous publications (Huang et al., 2019; Shen et al., 2019). Briefly, the data we use here are from the 6-week (26 July to 31 August 2016) deployment of a laser ablation aerosol particle time-of-flight mass spectrometer (LAAPTOF; AeroMegt GmbH) and a high-resolution time-of-flight aerosol mass

spectrometer (HR-ToF-AMS; Aerodyne Inc.; hereafter AMS). These instruments were used to provide real time information on size and chemical composition for individual particles and bulk samples, respectively. The LAAPTOF is a commercially available single particle mass spectrometer and has been well described in recent publications (Gemayel et al., 2016; Marsden et al., 2016; Ramisetty et al., 2018; Shen et al., 2018). In brief, aerosol particles with a size range of 70 nm to 2.5 μm vacuum aerodynamic diameter ($d_{va}$) are sampled with a flowrate of ~80 cm$^3$ min$^{-1}$, focused and accelerated via an aerodynamic lens

(ADL). Afterwards, the particles of ~200 nm to 2.5 μm $d_{va}$ can be detected with two laser diodes (λ = 405 nm) mounted in the

particle time-of-flight (PTOF) chamber. Once a single particle is detected successively by both of the detection lasers, its $d_{va}$ is calculated from the time of flight and an excimer laser ($\lambda$ = 193 nm) is triggered to desorb and ionize the refractory and non-refractory species of the particle. The resulting cations and anions are analysed with a bipolar ToF mass spectrometer. The AMS measures the particle size-resolved bulk chemical composition, e.g., organic compounds, ammonium, nitrate, sulfate, and chloride, in mass concentrations with high time resolution and sensitivity (DeCarlo et al., 2006). Similarly as the LAAPTOF, aerosols are sampled with a flowrate of ~84 cm$^3$ min$^{-1}$ via a similar ADL (Peck et al., 2016) in the size range 70 nm to 2.5 µm $d_{va}$ and then pass through a PTOF chamber. At the end of the PTOF chamber, particles encounter a 600 ℃ heater that vaporises the non-refractory species. The corresponding vapours are ionized by electron impact with high energy of 70 eV, generating positive ions that are analysed with a ToF mass spectrometer.

All sampling inlets (PM$_{2.5}$, PM$_{10}$, and total suspended particulates) were positioned at the top of a measurement container (3.7 m above ground level, a.g.l.). Particle number (size>2.5 and >7 nm, CPC3776 and CPC 3022A, TSI Inc.), particle mass (PM$_{2.5}$, PM$_{10}$, and total suspended particulates; FIDAS, PALAS GmbH), black carbon (AE33-7 Aethalometer, Magee Scientific Co.), and trace gases O$_3$ (O341M, Environment SA), SO$_2$ (AF22M, Environment SA), NO$_2$ (AS32M, Environment SA) CO$_2$ (NGA2000, Rosemont Inc.) were measured. Meteorological parameters temperature, relative humidity, solar radiation, wind direction and speed (WS700, Luft GmbH) were measured 10 m a.g.l. and complemented by meteorological data from a 200 m tower located at the KIT campus north, ~2 km east of the measurement site (49°5′43.0″N 8°25′46.0″E).

### 2.2 Mass spectrometer data analysis

Single particle data were analysed by using the Igor LAAPTOF data analysis software (version 1.0.2, AeroMegt GmbH). The corresponding basic procedures (e.g., mass calibration and fuzzy c-means clustering) and particle mass quantification method have been described in detail in our previous studies (Shen et al., 2018 and 2019). The AMS data analysis was done with the standard software written for Igor, including SQUIRREL (version 1.60C) and PIKA (version 1.20C). To account for a particle bouncing effect, we applied a composition-dependent collection efficiency (CE, the product of net particle transmission and detection efficiency) to the entire AMS dataset (Canagaratna et al., 2007; Middlebrook et al., 2012). In order to investigate the potential sources for organic aerosols, we have done positive matrix factorization (PMF) analysis for AMS high-resolution (HR) spectra of organic compounds, by using the PMF Evaluation Tool (PET version 3.00D) (Ulbrich et al., 2009). Detailed protocols were reported by Ulbrich et al. (2009) and (DeCarlo et al., 2010). In brief, the HR input matrices without isotopes were created using PIKA. A five-factor-PMF solution with FPEAK=0 was chosen according to the following criteria, including least residuals (unexplained data), $Q/Q_{exp} \approx 1$ (quality-of-fit parameter Q is the sum of the squares of the scaled residuals; $Q_{exp}$ is the expected Q), and distinguishable features (e.g., spectra and diurnal patterns) of the components. It should be noted again that although fuzzy clustering and PMF analyses are commonly used methods in source apportionment, they cannot provide the information of aerosols origin and the interaction between aerosol transport and transformation processes, which can be studied with aerosol transport models.

## 2.3 Regional transport model COSMO-ART

The regional transport model COSMO-ART is used in this study to simulate an episode with high mass concentration of organics (up to ~25 µg m$^{-3}$ measured by AMS). Detailed model descriptions can be found in previous studies (Bangert et al., 2011; Knote et al., 2011; Stanelle et al., 2010; Vogel et al., 2009). Therefore, we will only give a brief review here. COSMO-ART is a fully online coupled model system, which uses the same spatial and temporal resolution for all scalars, such as temperature, gas and aerosol particle concentrations. The meteorological module is based on the operational weather forecast model COSMO of the German weather service (Deutscher Wetterdienst, DWD). When atmospheric chemistry and aerosol dynamics are coupled to COSMO, it allows the online calculation of the transport of reactive trace gases and aerosol particles as well as their interaction with the state of the atmosphere. Physical processes, e.g., nucleation, coagulation, condensation and sedimentation are included in the model. The aerosol particles are represented by five interactive lognormal modes representing particles consisting of internally mixed sulfate, ammonium, nitrate and organic compounds, pure soot, as well as internal mixtures of them all. The hourly biogenic emissions are calculated online according to the land use data, the simulated temperatures, and radiative fluxes (Vogel et al., 1995). The anthropogenic emissions, including traffic emissions, large point and area sources such as industrial and households' areas, were pre-calculated by J. Bieser at Helmholtz Zentrum Geesthacht (Backes et al., 2016; Bieser et al., 2011). In our study, the simulation domain is covered by a 350×330 grid with a horizontal resolution of 1×1 km$^2$ and 50 vertical atmospheric layers (nonlinear increase up to 22 km a.g.l.). Most of the simulation results shown in this paper correspond to the lowest atmospheric level (10 m a.g.l.) and we indicate if results are from higher levels. The hourly emissions of $SO_2$, $NO_x$, CO, $NH_3$, anthropogenic volatile organic compounds (VOCs), and primary organic aerosols (POA) were pre-calculated. As shown in Fig. S1, VOCs, $NO_x$ and POA are mainly emitted at a lower atmospheric level in urban areas, such as the cities of Karlsruhe, Stuttgart, Mannheim, and Strasburg, as well as the corresponding highways. $SO_2$ hot spots typically correspond to the stack heights of the power plants emitting, such as the one in the western part of Karlsruhe.

## 3 Results and Discussion

In the first section (3.1), we give an overview of the meteorological conditions, the trace gases, and particles measured during the whole measurement campaign and will then give an example for source identification using the transport model in section 3.2. In the following two sections (3.3 and 3.4), we discuss two selected episodes in more detail, which are dominated by aged sea salt particles and relatively high organic particle mass concentrations, respectively.

## 3.1 Overview of the field observation

Table 1 and Fig. S2 give an overview of meteorological conditions, trace gases, and particle concentrations for the entire measurement campaign. In summer 2016, the major wind direction was southwest $(174 \pm 75)°$ and an average wind speed was $(1.3 \pm 0.9)$ m s$^{-1}$ at the measurement site in the upper Rhine valley. This wind speed allows trace gases and particles to be transported over a distance of 10 km within 2 hours e.g. from the city of Karlsruhe to the measurement site. During the

measurement time the sunrise and sunset were at around 06:30 and 20:30, respectively. Temperature and relative humidity (RH) were showing relatively stable anti-correlated diurnal patterns. Given the frequent clear nights, night-time temperature inversions occurred for 32 out of 37 nights during the entire measurement period (Fig. S3). Among the trace gases, $CO_2$ and $O_3$ were showing clear diurnal patterns with ozone peaking in the afternoon. $NO_2$ concentrations were higher in the morning hours and in the late evenings and generally were anti-correlated to $O_3$. $NO_2$ and particle number were showing spikes at similar times (more frequently during the day-time), mainly due to the local emissions from nearby road traffic. The averaged particle number concentration of $(1.2 \pm 0.8) \times 10^4 \, cm^{-3}$ for particles larger than 2.5 nm measured by CPC3776 in this study, is comparable with that observed at the same site in previous tram-based measurements (Hagemann et al., 2014). $SO_2$ concentrations showed peaks with low frequency (up to >10 ppb) but only when the wind was coming from the direction of the industrial area in the southwest. As shown in Fig. S2, these $SO_2$ peaks correspond to higher particle number concentrations at around noontime, when the wind speed is higher (~1.7 m s$^{-1}$) and vertical mixing intensified. This is in agreement with downward mixing of stack emissions from industrial sources, which can be validated by model calculations using COSMO-ART (refer to section 3.2).

In the entire measurement period, the total organic compounds, sulfate, nitrate, and ammonium were analysed to contribute on average $(58 \pm 12)$ %, $(22 \pm 7)$ %, $(10 \pm 1)$ %, and $(9 \pm 3)$ % to the total non-refractory $PM_{2.5}$ mass measured by AMS, respectively. Given the AMS-PMF results shown in Fig. S4, five OA factors (components) were determined with characteristic mass spectra and corresponding elemental ratios (O:C and H:C), and distinct temporal variations. There were $(9 \pm 7)$ % HOA, $(16 \pm 11)$ % SV-OOA, and $(75 \pm 15)$ % LV-OOA, i.e., $(16 \pm 15)$ % LV-OOA1, $(37 \pm 17)$ % LV-OOA2, and $(21 \pm 13)$ % LV-OOA3, contributing to the OA mass. The HOA and OOA are linked to primary and secondary organic aerosol (POA and SOA), respectively (Ulbrich et al., 2009). Thus, the SOA dominated the OA mass at this measurement site. Further descriptions of PMF results can be found in the supplement (Fig. S4) and more discussions are given in section 3.4.2 especially regarding the organic compounds.

In a previous publication (Shen et al., 2019), we could identify seven major particle classes at the measurement site among ~$3.7 \times 10^5$ single particles with sizes between 200 nm and 2.5 μm $d_{va}$. In brief, Class 1 are calcium-rich and soil-dust-like particles (calcium-soil) with marker peaks at m/z 40 Ca$^+$, 56 CaO$^+$, 57 CaOH$^+$, 75 CaCl$^+$, 96 Ca$_2$O$^+$, and 112 (CaO)$_2^+$. This particle class shows a good correlation (Pearson's correlation coefficient $\gamma \geq 0.6$) with the laboratory derived reference spectra of soil dust particles. Class 2 are aged-soot-like particles (aged soot) with characteristic $C_n^\pm$ patterns and are mixed with sulfate and nitrate. Class 3 are sodium-salt-like particles (sodium salts) with marker peaks at m/z 23 Na$^+$, 81/83 Na$_2$Cl$^+$, 139/141 Na$_3$Cl$_2^+$, 165 Na$_3$SO$_4^+$, 35/37 Cl$^-$, 93/95 NaCl$_2^-$, 115 Na(NO$_2$)$_2^-$, 131 NaNO$_2$NO$_3^-$, and 147 Na(NO$_3$)$_2^-$. Class 4 are secondary-inorganics-rich and amine-containing particles (Secondary inorganic–amine) with prominent ammonium, nitrate and sulfate markers, as well as amine marker peaks at m/z 58 C$_2$H$_5$NHCH$_2^+$, 59 (CH$_3$)$_3$N$^+$, 86 (C$_2$H$_5$)$_2$NCH$_2^+$, 88 (C$_2$H$_5$)$_2$NO/C$_3$H$_6$NO$_2^+$, and 118 (C$_2$H$_5$)$_2$NCH$_2^+$. Class 5, aged-biomass-burning and soil-dust like particles (biomass burning–soil), has aged biomass burning marker peaks at m/z 39 K$^+$, 213 K$_3$SO$_4^+$, 46 NO$_2^-$, 62 NO$_3^-$, 97 HSO$_4^-$, and peaks related to aromatic compounds (50 C$_4$H$_2^+$, 63 C$_5$H$_3^+$, 77 C$_6$H$_5^+$, 85 C$_7$H$^+$, 91 C$_7$H$_7^+$, 95 C$_7$H$_{11}^+$, 104 C$_8$H$_8^+$, 115 C$_9$H$_7^+$). Furthermore, class 5 shows a strong correlation with reference spectra of soil dust particles. The representative spectra of class 6, aged-biomass-burning and organosulfate-containing

particles (biomass burning–organosulfate), also features aged biomass burning particles and has organosulfates marker peaks at m/z 141 $C_2H_5OSO_4^-$, 155 $C_2H_3O_2SO_4^-$, and 215 $C_5H_{11}OSO_4^-$. Class 7 particles contain all the marker peaks from the other six classes and show a good correlation with the reference spectra of mineral and soil dust particles, and are consequently labelled as "mixed/aged dust" (short for mixed/aged and dust-like particles). These particle types were not always present in similar fractions but showed distinct periods with one or two types dominating the particle mass (Shen et al., 2019). Here, we will mainly discuss two particle classes that dominated the total particle mass, i.e., sodium salts and biomass burning–soil particles, contributing on average (36 ± 27) % and (30 ± 18) % to the PM$_{2.5}$ mass measured by the LAAPTOF, respectively. We will also discuss one characteristic particle class, i.e., secondary inorganic–amine particles, which showed relatively intensive and distinct peaks three times during this campaign (Fig. S2) as well as several weaker peaks.

As shown in Fig. 2 (a), total particle mass concentrations have clear diurnal pattern (the data is averaged over the entire campaign), showing higher night-time values and a peak during the morning rush hour (~9:00 local time). Higher particle mass values mainly correspond to southern wind directions, with one exception of extremely high masses associated with north-westerly winds (Fig. 2c). Specifically, sodium salt containing particles have higher fractions during day-time and correspond to south-westerly wind directions (Fig. 2b). Such wind was observed mainly during day-time and with higher wind speeds than at night. Biomass burning–soil particles enrich during night-time and mainly correspond to southeast wind (Fig. 2d). Secondary inorganic–amine particles typically rise in the morning and correspond to south and southwest wind directions (Fig. 2). When we compare weekdays (Monday to Friday) and weekends (Saturday and Sunday) in Fig. S5, the sodium salt particles were always correlated with southwest wind directions. Obvious differences can be found for secondary inorganic–amine and biomass burning–soil, respectively. For secondary inorganic–amine type particles, they were mainly advected by south and southwest wind during weekdays. However, they were much less abundant during weekends and did not show a clear relationship with wind directions (only a small peak for northwest). For biomass burning–soil particles the observations are a bit more complex.

The biomass burning–soil particles measured during weekdays were correlated with south and southwest wind as well, indicating potential sources from the city of Karlsruhe and its industrial area. Please note, that the representative spectra of these particles show aromatic ion peaks which may arise from aromatic compounds such as toluene and xylene due to motor vehicle emissions. Besides, this type of particles contains also some metals such as barium and lead that are related to coal combustion from industry (Hudson et al., 2004; Xu et al., 2018). These features correspond to urban and industrial sources. During the weekends, biomass burning–soil particles had higher mass concentrations than during weekdays and were correlated with wind directions from east, southeast and northwest, where are mainly forests and agricultural areas. Therefore, we can infer that there were various sources for biomass burning–soil particles and these sources differed between weekdays and weekends. This could be e.g., barbeque fires during weekends while the weekdays' source is likely to be associated with that for secondary inorganic–amine particles.

In order to refine this type of particles, we compared them with black carbon particles measured by an Aethalometer (AE33-7). The equivalent black carbon concentration was calculated by the mass absorption cross sections value of 11.7 $m^2\,g^{-1}$ (Herich

et al., 2011). They seem to show a similar trend in the time series with a triplet of peaks from August 25[th] to 28[th] (refer to Fig. S2 and Fig. S6). We found that biomass burning-soil and black carbon concentrations are well correlated only for the weekends, with Pearson's correlation coefficients $\gamma \geq 0.6$ (Fig. S6 b), and poorly correlated for weekdays ($\gamma < 0.3$). This points to different major sources for weekdays and weekends. In addition, the black carbon instrument measuring the light absorption at different wavelengths ($\lambda$) can be used to determine the absorption Ångström exponent ($\alpha$) (Drinovec et al., 2015). For the entire campaign, the average $\alpha$ was $0.96 \pm 0.24$, calculated by using a pair of wavelengths and their corresponding mass absorption cross sections (MAC=11.7 and 23.6 $m^2g^{-1}$ for $\lambda$=950 and 470 nm, respectively, Herich et al., 2011). Herich et al. (2011) and Zotter et al. (2017) have assigned an $\alpha$ = ~0.9 to BC from road traffic emissions, while the $\alpha$ values for wood burning BC were typically higher ~1.68 (Zotter et al., 2017). Our measurements indicate that traffic was the dominating source of BC. Considering the good correlation between BC and biomass burning–soil particles during the weekend, we could infer the corresponding traffic emissions that contributed to biomass burning–soil particles. All observations support the conclusion that this particle class has significant anthropogenic sources, which may vary between weekends and weekdays. It should be noted that, there were no significant biomass burning organic aerosols (BBOA) identified by AMS-PMF for the entire measurement period. The marker f60 (the mass fraction of m/z 60 to the total organics) of BBOA being very close to the background fraction of 0.003 (Crippa et al., 2014; Cubison et al., 2011). In addition to biomass burning, also other potential sources for potassium and sulfate rich particles detected by LAAPTOF must be taken into consideration, such as coal combustion (Xu et al., 2018). Therefore, further analysis is required to identify the different sources contributing to this particle class.

As mentioned before, the dominating particle classes varied for different time periods (Shen et al., 2019). In the following we focus on an episode with particles rich in sodium salts, to find out if sea salt particles can significantly influence the aerosol particle composition at this central European location about 800 km inland. Furthermore, we discuss another episode with relatively high organic mass concentrations to find out how these mass loads could form and exist for a relatively long time period. Both episodes are easily recognized from the time series of particles measured by LAAPTOF and the organic compounds measured by AMS (Fig. S2 and Fig. 3; see also Fig. 4 in Shen et al, 2019), as well as the time series of OOA, in particular for SV-OOA, LV-OOA1 and LV-OOA2 (Fig. S4). Apart from the difference of dominating particle classes and particulate organics, the particle acidity indicator value, organonitrate fraction and organic carbon oxidation state, as well as meteorological conditions were quite different between these two episodes. For particle acidity, the molar ratio of $NH_4^+$ measured/predicted from AMS measurement, is commonly used as an indicator (Zhang et al., 2007). The predicted $NH_4^+$ value is the mass concentration required to fully neutralize sulfate, nitrate, and chloride measured by AMS. Lower ratio value represents higher acidity under certain conditions, i.e., inorganic species are dominating the particle mass and crustal elements have relatively low concentrations (Zhang et al., 2007). It should be noted that the organic acids and crustal elements such as sodium can bias the results (Hennigan et al., 2015, and the reference therein). Despite of the large uncertainty for indicating acidity, we still can use such ratio to indicate special events, e.g., sodium nitrate rich particles will provide nitrate, leading to lower values. As shown in Fig. 3, this ratio reaches lowest value during the sodium salts rich episode, while the value in the organic rich episode is at the average level. If we assume the organic acids only have minor influence on the particle acidity, the anti-correlation between

acidity indicator value and organic concentrations may be related to acid-catalysed organic particle formation. For the fraction of organonitrates, the AMS derived ratio of $NO^+/NO_2^+$ is commonly used as an indicator (Farmer et al., 2010; Kiendler-Scharr et al., 2016). In the episode rich in sodium salts, particles contain the highest organonitrate fraction while the particles in the organic rich episode contain the lowest fraction (refer to Fig. S7). It should be noted that $NaNO_3$ can also result in higher $NO^+/NO_2^+$ ratios (5 to 7, Allan et al., 2004) than $NH_4NO_3$ (2.3 in our calibration). However, the $NO^+/NO_2^+$ values here are much higher (c.f. Fig. S7) and thus we can expect a major contribution by organonitrates. Higher fraction of organonitrates suggest a stronger influence of anthropogenic emissions on the sodium rich aerosol. The averaged carbon oxidation state ($\overline{OS}_C$) of the organic compounds is a robust metric of its degree of oxidation or processing (Canagaratna et al., 2015; Kroll et al., 2011). It turns out that the organics were more oxidized in the organic rich episode with relatively higher $\overline{OS}_C$ value and less variability (-0.10 ± 0.20) than that for the entire campaign (-0.19 ± 0.23) (refer to Fig. S7 and Table 2). In addition, the major wind with higher speeds was from southwest in the episode rich in sodium salt particles, and we will show in section 3.3 that this is related to long-range transport of sea salt particles. For the organic rich episode, the wind speeds were lower for most of the time. The different features of these two episodes are summarized in Table 2. We will discuss the episodes in more detail in sections 3.3 and 3.4, but will first demonstrate the usefulness of the transport model to identify an industrial point source.

## 3.2 Identification of an industrial point source

As shown in Fig. 4, we have modelled $O_3$, $NO_2$, $SO_2$, and particle number for the organic rich episode with COSMO-ART. Model simulations show comparable results and/or trends with field observations. For $SO_2$, there is a modelled peak at 10:00 on August 26th. As mentioned above, in the upwind direction of our measurement site is a coal fired power plant, where $SO_2$ is emitted from a stack at 230 m a.g.l.. Therefore, we compared $SO_2$ concentrations in an area including the power plant and our measurement site (refer to Fig. 5 a1 and a2), as well as time series of their vertical profile at the measurement site (Fig. 5b). The model results show that $SO_2$ emissions were starting at 02:00 from the stack of the power plant, continuously transported by southwest wind (from 06:00, not shown here). Around noon, the $SO_2$ was transported downward due to enhanced vertical mixing. Similar to $SO_2$, we found corresponding vertical transport of $NO_x$ and primary particles (refer to Fig. 5 c and d), as well as secondary inorganic particles which will be discussed in section 3.3.3. In addition to the power plant emissions, particle emissions from the Karlsruhe refinery (e.g., around 8:00 at 122 m a.g.l.) and downtown Karlsruhe (at 10 m a.g.l.) were observed in the model results, which are visualized via videos of the evolution of particle number at these two altitudes (refer to Data availability). Given these findings, we have verified industrial contributions to the particles measured at this rural site. However, the particle numbers calculated by the model are 10 times lower than the measured values. In this context, it should be noted that, the observations relate to particles larger than 2.5 nm and 7 nm, respectively. The modelled particles include Aitken and accumulation mode particles with median diameter at around 30−80 nm and 300−400 nm, respectively. The model does not represent the ultrafine particles to their full extend, especially e.g., the contribution of organic compounds or ammonia to nucleation.

### 3.3 Aged sea salt in central Europe

Sodium salt particles accounted on average for $(35.6 \pm 26.7)$ % of the particle mass measured by LAAPTOF during the whole campaign, and they dominated several time periods (refer to Fig. S2). However, not all the peak events were the same. We chose a unique episode when sodium chloride signatures, e.g., m/z 81 $Na_2Cl^+$, were showing the strongest signal, as plotted in Fig. 6 (a). In this episode, the sodium salts accounted for $(70 \pm 24)$ % of total mass of the single particles. To study the origin of these particles, we used the Hybrid Single-Particle Lagrangian Integrated Trajectory (HYSPLIT; Stein et al., 2015) on the READY website (http://www.ready.noaa.gov; Rolph et al., 2017) for back-trajectory analysis. The corresponding result suggests that these particles were coming from the Atlantic Ocean, which is more than 800 km west of our measurement site (refer to Fig. S8 a). Therefore, we can constrain the type of sodium salts particles as sea salts with an age of about 1.5 to 3 days.

In this episode, sea salt particles retained strong chloride signatures in spite of the long-range transport, due to relatively high wind speeds (e.g., the average wind speed was ~6 m s$^{-1}$ from the ocean to our measurement site at noontime on August 21$^{st}$ and 22$^{nd}$, based on 72 h back-trajectory analysis). Therefore, we interpret them as relatively fresh sea salt, which is the 1$^{st}$ sub-class of sodium salts particles identified with the fuzzy clustering analysis of the single particle mass spectra. As shown in Fig. 6 (a), there is an obvious decrease of m/z 81 $Na_2Cl^+$ at the end of this episode, while the sodium nitrate signature at m/z 108 $Na_2NO_3^+$ increases. Such trends can also be found in similar but much less prominent sodium chloride rich episodes (see the red guiding arrows in Fig. 6a). This indicates the aging process of sea salt particles, namely heterogeneous replacement of chloride by nitrate on sea salt particles (Gard et al., 1998). We assigned the 2$^{nd}$ sub-class of sodium salts to more aged sea salt particles, which are bigger than the relatively young sea salt (Fig. 6b). 72 h back-trajectory analysis suggests that these particles were coming from the Atlantic Ocean and were carried with south-westerly wind of relatively lower speed (~3.5 m s$^{-1}$) to reach our measurement site on August 23$^{rd}$, thus they spent more time (~3 days) over the continent and are more aged. The representative spectra of sodium salt particles and the averaged spectra of their sub-classes can be found in Fig. S9. The signal difference between aged and young sea salts spectra (aged - young) is shown in Fig. 6 (c), highlighting their signatures. Similar relationships as for m/z 81 $Na_2Cl^+$ and 108 $Na_2NO_3^+$ can also be found for the other signature peaks (refer to Fig. S10). It is worth noting that there are also nitrate related peaks in young sea salt, such as m/z 104 $NaClNO_2^-$, verifying that these young sea salts were not fresh. In addition, m/z 92 $Na_2NO_2^+$ and 131 $NaNO_2NO_3^-$ increase earlier than 108 $Na_2NO_3^+$ and 147 $Na(NO_3)_2^-$. Such that, we can infer the aging degree from a series of signature peaks in LAAPTOF spectra: 81 $Na_2Cl^+$ →104 $NaClNO_2^-$ → 92 $Na_2NO_2^+$ and 131 $NaNO_2NO_3^-$ → 108 $Na_2NO_3^+$ and 147 $Na(NO_3)_2^-$ (from relatively fresh to more aged, corresponding to chloride to nitrite to nitrate salts). However, this requires further systematic studies to be validated.

For sodium salt particles, a weak but characteristic peak at m/z 129 $C_5H_7NO_3^+$ is likely to be an organonitrate signature (Shen et al., 2019). As shown in Fig. S10, m/z 129 $C_5H_7NO_3^+$ has a similar trend as m/z 131 $NaNO_2NO_3^-$ and it is more pronounced in the LAAPTOF spectra of aged sea salt. In the sodium chloride rich episode, in general m/z 129 $C_5H_7NO_3^+$ has a similar trend as organonitrate mass concentrations estimated by AMS results and the results of a molecular analysis using FIGAERO-HR-ToF-CIMS (stands for a filter inlet for gases and aerosols coupled to a high-resolution time-of-flight chemical

ionization mass spectrometer) in a companion study by (Huang et al., 2019). The method for organonitrate estimation can be referred to Farmer et al. (2010) and the parameters used here are the same as those used by Huang et al. (2019). We have briefly described this method in the supplement after the caption for Fig. S10. In addition, our simulation chamber study has shown that this m/z 129 $C_5H_7NO_3^+$ arises from the recombination of inorganic nitrate and organics due to the laser desorption and ionization (LDI) matrix effect in single-particle MS (Murphy, 2007; Reilly et al., 2000; Reinard and Johnston, 2008; Zenobi and Knochenmuss, 1998) and even stronger when it is arising from organonitrates formed under $NO_x$ rich conditions (refer to Fig. S11). Taken together, we can conclude that the m/z 129 $C_5H_7NO_3^+$ signal is mainly resulting from organonitrates, but we still cannot rule out the LDI effect. Furthermore, our simulation chamber studies also revealed that m/z 129 $C_5H_7NO_3^+$ in LAAPTOF spectra could be a signature for anthropogenic sources (refer to Fig. S11 and the corresponding detailed description in the supporting information). Therefore, m/z 129 $C_5H_7NO_3^+$ on sea salt particles indicates interactions between the natural sea salt and anthropogenic emissions. It is worth noting that another weak but characteristic peak was observed at m/z 165 $Na_3SO_4^+$. This signature peak for sea salt sulfate can be found with stronger signal in young sea salt than in aged sea salt (Gard et al., 1998).

Apart from that, in high resolution AMS spectra we have observed the peak at 78.99 $CH_3SO_2^+$, which is the marker for methanesulfonic acid (MSA) from marine or coastal areas (Huang et al., 2015), although other sources cannot be excluded (Daellenbach et al., 2017). Based on the method developed by Huang et al. (2015), we have also estimated the MSA fraction ($f_{MSA}$, signal fraction of MSA to total org and sulfate) and mass concentrations for the entire campaign (refer to Fig. S12). It turns out that during the sodium chloride rich episode, the $f_{MSA}$ is enhanced compared to the other low sodium salts time period, such as organic rich episode. This also supports that sodium salts particles measured by the LAAPTOF had sea salt origin.

## 3.4 Origin of the aerosol particles during the organics rich episode

At the end of the measurement period, we observed three subsequent organic peaks (see Fig. 3, Fig. S2, or Fig. S7): one peak during Friday August 26[th] day-time and two peaks during following two nights. Organics measured in this episode exhibited the highest mass concentrations of $(12.9 \pm 3.9)$ µg m[-3], which accounted for $(77 \pm 6)$ % of the total non-refractory particle mass and is almost 3 times more than organic masses measured in other periods of the entire AMS measurement period. Please note, some secondary inorganic components were also measured in this episode, showing some peaks at similar times as the high organic peaks. As for the sodium chloride rich episode, we have also done a HYSPLIT back-trajectory analysis for the organic rich episode. Fig. S8 (b) shows these results: the air mass was coming from the continent during this episode.

In order to better interpret the measurements and to understand the impact of transport processes for the particles measured in the organics rich episode, we have done COSMO-ART simulations covering the time period from August 25[th] to 28[th], 2016. Fig. 7 shows the particulate organic compounds, the secondary inorganics (ammonium, nitrate, and sulfate) measured by AMS, and the result calculated using COSMO-ART (panels d to g). The corresponding meteorological conditions measured are shown in panels (a) to (c) of Fig. 7. In general, the trends and also the amplitudes of the model results compare quite well with the observations. In the following, we will combine field observation and transport model simulation results to address these

questions: 1) were the organic rich aerosols due to local formation, regional transport, or vertical mixing; 2) were all three organic peaks caused by the same reasons; 3) were the secondary inorganics originating from the same sources as the organics. We will start the discussion with the inorganics, will then go to the organics, and will end up with a typical particle type in this episode.

### 3.4.1 Secondary inorganic compounds

As shown in Fig. 7 (e) to (g), observations and model results for secondary inorganic compounds show comparable diurnal cycles and even the concentration levels match quite well. While ammonium and nitrate have a similar trend for most of the time, the trend for sulfate seems to be somewhat different. The major peaks of ammonium and nitrate appear at similar times as the high organic peaks. As will be shown in the following section, these compounds originated from anthropogenic sources in the urban and industrial area of Karlsruhe and from the city of Stuttgart. There is only one instant when all the measured secondary inorganics are showing peaks together: the inorganics are peaking at ~10:00 on August 26[th], when also high $SO_2$ concentrations were observed (c.f. Fig. 4) which were due to the emissions from a power plant located in upwind direction. To verify this, we can refer to the time series of vertical profiles for the simulated secondary inorganics (Fig. S13 b to d). Despite similar spatial distributions of nitrate, ammonium, $NO_2$, and $SO_2$ from 02:00 to 12:00 (c.f. Fig. 5) the measured sulfate peak for that time is missing in the model output. The reason for this is still unclear. It should be noted that, compared to the sulfate peak observed, the companion ammonium and nitrate peaks are much broader, attributed to additional anthropogenic sources from the city and refinery rather than only the power plant.

Some of the differences between model result and observation, such as that on August 25[th] or e.g., for sharper peaks, may be explained by to missing (point) sources in the emission data and/or the limited spatial resolution. The general agreement is however quite well and also for some point sources like the largest power plant in Karlsruhe. Some of the differences in the trends of nitrate and sulfate may be caused by the different nature of the sources of their precursor gases $SO_2$ and $NO_x$. While $SO_2$ is emitted in this area mainly from a few point sources, $NO_x$ is emitted from various sources over a larger area.

### 3.4.2 Organic compounds

The dominating OA components in this episode were LV-OOA1 (42 ± 15) % and LV-OOA2 (32 ± 21) %, which are surrogates for more aged SOA (Zhang et al., 2011). It should be noted that LV-OOA1 has strong correlation with SV-OOA ($\gamma = 0.8$), which is a surrogate for local (less-aged) SOA (Zhang et al., 2011). As shown in Fig. S4 (the insert in panel d), they are both peaking at a similar time during the night with broader peaks for LV-OOA1. This indicates that the LV-OOA1 might have a similar origin as the SV-OOA (more related with local biogenic sources). However, it can also be influenced by regional transport. For example, the first LV-OOA1 peak has two sub-peaks: the first one appears during the night, e.g., related to $NO_3$ oxidation (strong correlation with $\gamma = 0.9$ between LV-OOA1 and CHO1N family has been found here, refer to their time series in Fig. S4 b); and the second sub-peak appears in the morning rush hour and corresponds to south and southwest wind directions, indicating the potential transport from the urban and industrial area of Karlsruhe. The more oxidized LV-OOA2 is peaking

during day-time and shows a negative correlation with SV-OOA ($\gamma$ = -0.5), possibly due to photochemical aging of SV-OOA. Therefore, we conclude that the SOA measured in this study was likely due to both local formation and regional transport.

For the field and model comparison, the general temporal development of organics is represented well by the model, showing similar trends as measured by AMS, but underestimates the mass concentrations by a factor of ~2.3; averaged model result is (5.5 ± 3.3) µg m$^{-3}$, while observed were (12.9 ± 3.9) µg m$^{-3}$. Potential reasons for this discrepancy will be discussed in the light of sensitivity tests at the end of this section.

As mentioned above, the organics shows a peak on Friday August 26$^{th}$ during day-time (peak 1) and two more during the following nights of the weekend (peak 2 and 3) in the organics rich episode. Model result cover peak 1 and 2, as well as of the first part of peak 3. Analysis of the model results shows that the three peaks were caused by different reasons including meteorological conditions, regional transport, and local or in-situ new particle formation and growths (such as night-time new particle formation; Huang et al., 2019). As shown in Fig. 7a and d, peak 1 corresponds to south and southwest wind, indicating that the air masses came from the urban and industrial area of Karlsruhe. This is illustrated very well in a video showing the transmission of particulate organics leading to peak 1 (refer to Data availability). The most intensive peak in organic particle mass (peak 2), starts from August 26$^{th}$ 20:30 (Fig. 3 and Fig. S2) when the wind direction shifts from northwest to east (northeast, straight east, and southeast), the wind speed decreases to < 0.5 m s$^{-1}$, and the night-time temperature inversion starts. Such stagnant conditions limit aerosol mixing and support build up aerosols and their growth. During the entire campaign, the strongest temperature inversion was observed during this night from August 26$^{th}$ to 27$^{th}$ (peak inversion at 02:40 on August 27; Fig. S3). In spite of the stagnant conditions and rather low wind speeds, regional transport still plays an important role. The model results for this time show transport of organic aerosol particles from the city of Stuttgart (southeast to the measurement site) over forested areas into the Rhine valley. This is clearly visible in a corresponding video of the model results for particulate organics and wind directions (refer to Data availability). Furthermore, we investigate the relative contribution of biogenic and anthropogenic emissions and their interactions for this episode by comparing the model results with and without biogenic emissions. It turns out that the corresponding air masses originating from anthropogenic emissions in the Stuttgart area were transported to the measurement site by south-easterly winds via forested hills. During this transport, the organic particle mass concentrations increased while being mixed with biogenic emissions from the forest. The resulting air mass finally reaching the measurement site consisted therefore of anthropogenic and biogenic organics. For peak 3, we can also find impact of transport as illustrated in another video of particulate organic mass evolution (refer to data availability). However, the origin of this peak is more complex and is most likely more influenced by local particle formation and growths. After the night-time peak 2, from 12:00 on August 27 the air mass were doing back and forth movements in west-east directions, showing a strong concentration gradient (west-east). Such conditions lead to a high uncertainty of the model results for one point. Hence, although we observed quite similar particle compositions in these three subsequent peaks with high organic mass concentrations for similar meteorological conditions with low wind speeds, all three peaks were caused by air masses of different origin.

As mentioned above, the model underestimates the organic aerosol mass on average by a factor of 2.3. In order to better understand this discrepancy, we have done three sensitivity tests:one without biogenic emissions (test 01), one with doubled

biogenic emissions (test 02) and one with boundary values or regional background values (outside the model domain) of particulate organic compounds increased by a factor of 10 (test 03). The results of test 01 suggest that $(80 \pm 4)$ % of the organic mass calculated was resulting from biogenic sources for the peak 1 time period, corresponding to the air masses coming from Karlsruhe. Regarding the time period of peak 2, there was a sharp drop of the biogenic contribution from the 88 % to 68 % corresponding to August 26[th] 23:00 to August 27[th] 2:00 when the air mass was coming from direction of Stuttgart (see Fig. 8). Hence, we can infer that the anthropogenic contribution to the air mass that originated from Stuttgart was somewhat larger than that originated from Karlsruhe. Over the entire organic rich episode, on average $(74 \pm 7)$ % of the organics mass was contributed by biogenic sources. To test if missing emissions of biogenic precursors can explain the underestimated organic particulate masses we doubled this major (biogenic) source for organics (test 02). As shown in Fig. 9, doubling biogenic emissions makes some of the peaks of organic mass fit the observations better but the concentrations before and in between the peak values are still underestimated. Another potential reason for the discrepancy could be to low boundary values (regional background); therefore, we increased them by a factor of 10 (test 03). This elevates the overall simulated concentration levels, leading to an overestimation especially of the first peak. These tests demonstrate the potential role of precursor emissions and regional transport of organic aerosol particles on the model results. There seems to be evidence that the model underestimates the regional background of organic aerosol particles. However, we cannot rule out that also underestimated sources of VOC or weaknesses in the SOA formation mechanism may contribute to the underestimation of the organic aerosol particle mass. For example, night-time new particle formations mechanisms are not included in the current model version.

## 4 Conclusions and atmospheric implication

High particle number concentrations have been observed at a rural site in the upper Rhine valley by measurements on a tramline (Hagemann et al., 2014). In order to study the nature of the aerosols in the same region, we conducted a 6-week stationary field measurement campaign at a rural site next to the tramline in summer 2016, and applied a regional transport model COSMO-ART to analyse the origin of the aerosols. Comparing the field observations and model calculations, we have verified that industrial emissions contribute to the high particle number concentrations at this rural site.

For the whole measurement period, the particles that dominate the mass are sodium salts, which contributed $(36 \pm 27)$ % to the $PM_{2.5}$ mass measured by the LAAPTOF. According to the AMS measurements, organics, sulfate, nitrate, and ammonium contributed $(58 \pm 12)$ %, $(22 \pm 7)$ %, $(10 \pm 1)$ %, and $(9 \pm 3)$ % to the total non-refractory $PM_{2.5}$ mass, respectively. AMS-PMF results suggest the organic aerosols (OA) consisted of $(9 \pm 7)$ % HOA, $(16 \pm 11)$ % SV-OOA, and $(75 \pm 15)$ % LV-OOA with three sub-types (LV-OOA1, LV-OOA2, and LV-OOA3). Furthermore, we have investigated two episodes, which were dominated by aged sea salt and organics, which show characteristic features such as higher particle acidities and carbon oxidation states, respectively. Both episodes selected, were dominated by particles formed from the interaction of natural/biogenic and anthropogenic emissions.

For the sodium chloride rich episode, sodium salt particles accounted for $(70 \pm 24)$ % of total mass of the single particles. We could show that the dominating particles were originally sea salt, from the Atlantic Ocean (~800 km away from the measurement site). These particles consisted of two classes with different aging stages. After $1.5-3$ days of transport, the original sea salt particles retained strong chloride signatures while after $3.5-4$ days of transport over the continent aged sea salt with stronger nitrate signatures and larger size arrived at the measurement location. In addition, the organonitrate signature peaks at m/z 129 $C_5H_7NO_3^+$ were stronger for the aged sea salt particles. According to chamber experiments on OH oxidation of $\alpha$-pinene and/ or toluene in the presence of $NO_x$ on sodium nitrate seed particles, we have evidence that m/z 129 $C_5H_7NO_3^+$ is a signature peak for anthropogenic organics in single particle mass spectra. Hence, the m/z 129 $C_5H_7NO_3^+$ observed on sea salt particles in this study is another indicator for the interaction of anthropogenic emissions with the sea salt particles. Besides, some other signature peaks such as 81 $Na_2Cl^+$, 92 $Na_2NO_2^+$, 108 $Na_2NO_3^+$, 104 $NaClNO_2^-$, 131 $NaNO_2NO_3^-$, and 147 $Na(NO_3)_2^-$ have the potential to indicate degree of aging of sea salt particles.

For the organics rich episode, the particulate organic compounds accounted for $(77 \pm 6)$ % of total non-refractory particle mass measured by the AMS. AMS-PMF analysis suggests the dominating OA components were LV-OOA1 and LV-OOA2, contributing $(42 \pm 15)$ % and $(32 \pm 21)$ % to the total OA mass in this episode. LV-OOA1 and LV-OOA2 had positive $(\gamma = 0.9)$ and negative $(\gamma = -0.5)$ correlation with SV-OOA, respectively, indicating their local formation. In order to investigate origin of the aerosols and their transport and transformation processes, we applied a regional transport model, COSMO-ART. Transport model results suggest that $(74 \pm 7)$ % of the particulate organic mass calculated was related to biogenic emissions. With the combination of mass spectrometry and COSMO-ART modelling, we can confirm that during Friday, August 26[th], air masses came from an industrial area and the city of Karlsruhe, leading to high particle numbers (including all the particles with size >2.5 nm), high particulate organics and secondary inorganics, as well as high fractions of aged-biomass-burning and soil-dust like particles. During night-time on August 26[th], south-easterly winds transported air masses from Stuttgart via forested areas to the measurement site, resulting in particles formed by the interaction of anthropogenic and biogenic precursors. Such air masses flowed into the Rhine valley and contributed also to the high organic aerosol mass concentrations during the night from August 27[th] to 28[th]. In addition, low wind speeds and temperature inversion limited mixing and supported new particle formation and growth, partially contributing to higher organic particle masses.

Comparing comprehensive field observations with the COSMO-ART model results showed quite good agreement for trends and most concentration levels for several aerosol particle compounds and trace gases even if not all details could be reproduced by the model. However, the simulated particle numbers are 10 times less than the observed values. This is mainly due to the fact that not all new particle formation processes (nucleation mode) are included in the model (e.g. night time chemistry as well as the role of organics or ammonia). Furthermore, the model underestimates particulate organics by a factor of 2.3. This discrepancy may be partly explained by an underestimated regional background, missing sources, or incomplete SOA formation and growths mechanisms, in the current model version. Such field-model comparisons and corresponding sensitivity tests are useful for targeting missing sources and mechanisms in the current model and thus help its development. Such that model can be improved to become better predictive, which is useful for e.g., legislation and implementation of policies against air pollution.

Consequently, model systems suitable for a reasonable environmental policy should be validated by comprehensive field studies with state-of-the-art aerosol analytics and meteorological parameters.

As discussed above, we can conclude that factors related to topography, metrological conditions, local emissions, in-situ formation and growths, regional transport, as well as the interaction of biogenic and anthropogenic compounds need to be considered for a comprehensive understanding of aerosol composition and processing in this region. Our results show how close

the physical and chemical nature of aerosols is related to emission sources, transformation processes, and complex transport pattern. The method of combination of field measurement and model calculation in this study can be widely used in other places in the world, in order to investigate the origin of aerosols and their interaction between transport and transformation. For example, in some pollution hot spots like Po Valley, and some other places which are subject to both biogenic and anthropogenic emissions. Besides, it is also not limited to the summer time that we studied. Taken together, this study provides a good insight

into the chemical nature and complex origin of aerosols in the upper Rhine valley, achieved by combining complimentary mass spectrometry and transport modelling.

**Data availability**

Five supporting videos (.gif files) are available at KIT open data (https://doi.org/10.5445/IR/1000094401).

**Author contributions**

X.S. operated LAAPTOF and AMS during the whole field campaign, did the LAAPTOF data analysis, did COSMO-ART model runs and model output data analysis, did HYSPLIT back-trajectory analysis, did further analysis for AMS data, produced all figures, and wrote the manuscript. H.S. organized the campaign, provided trace gas, meteorology, and particle measurements as well as suggestions for the data analysis, interpretation, and discussion. H.V. and B.V. provided the input data for COSMO-ART model, provided suggestions for the model simulation, the model data analysis, interpretation, and discussion. W.H.

operated AMS during the whole campaign and did basic AMS data analysis, as well as PMF analysis. C.M. helped to operate the instruments, did black carbon data analysis, and provided suggestions for the data analysis, interpretation, and discussion. R.R. helped to operate LAAPTOF. A.S.H.P. gave comments for interpretation of black carbon data. T.L. gave general advices and comments for this paper. All authors contributed to the final text.

**Competing interests**

The authors declare no conflict of interest.

## Acknowledgements

The authors gratefully thank the IMK-AAF AIDA staff and IMK-TRO modellers at KIT for helpful discussions and technical support, and the China Scholarship Council (CSC) for financial support of Xiaoli Shen and Wei Huang. Special thanks go to the Albtal-Verkehrs-Gesellschaft (AVG) for providing power and the measurement location near the tramline, to Norbert Kalthoff's group for providing tower data. Besides, the authors gratefully acknowledge the NOAA Air Resources Laboratory (ARL) for provision of the HYSPLIT transport and dispersion model and the READY website (http://www.ready.noaa.gov).

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

**Table 1: Mean values (±sdev) of characteristic parameters for the entire measurement period.**

| Parameter | Mean value ± sdev | Parameter | Mean value ± sdev |
|---|---|---|---|
| Wind Direction (°) | 174.0 ± 75.0 | **Particulate compounds measured by AMS** | |
| Wind Speed (m s$^{-1}$) | 1.3 ± 0.9 | Organics (µg m$^{-3}$) | 5.1 ± 3.3 |
| Temperature (K) | 294.4 ± 4.5 | Ammonium (µg m$^{-3}$) | 0.7 ± 0.4 |
| Relative Humidity (%) | 70.0 ± 17.0 | Nitrate (µg m$^{-3}$) | 0.8 ± 0.8 |
| $O_3$ (ppb) | 26.5 ± 15.8 | Sulfate (µg m$^{-3}$) | 1.7 ± 0.8 |
| $NO_2$ (ppb) | 11.4 ± 6.9 | **Seven particle classes measured by LAAPTOF** | |
| $SO_2$ (ppb) | 1.2 ± 2.7 | C1: Calcium-soil (µg m$^{-3}$) | 0.4 ± 0.5 |
| $CO_2$ (ppm) | 446.0 ± 32.0 | C2: Aged soot (µg m$^{-3}$) | 0.1 ± 0.4 |
| Particle Number ($10^4$ cm$^{-3}$) (size >2.5 nm) | 1.2 ± 0.8 | C3: Sodium salts (µg m$^{-3}$) | 1.2 ± 1.2 |
| Particle Number ($10^3$ cm$^{-3}$) (size: 2.5 to 7 nm) | 4.7 ± 4.9 | C4: Secondary inorganics-amine (µg m$^{-3}$) | 0.1 ± 0.6 |
| BC (µg m$^{-3}$) | 0.8 ± 0.7 | C5: Biomass burning–soil (µg m$^{-3}$) | 1.2 ± 1.3 |
| $PM_{2.5}$ (µg m$^{-3}$) | 6.3 ± 3.2 | C6: Biomass burning–organosulfate (µg m$^{-3}$) | 0.4 ± 0.6 |
| $PM_{10}$ (µg m$^{-3}$) | 8.6 ± 4.2 | C7: Mixed/aged dust (µg m$^{-3}$) | 0.5 ± 0.7 |

Note that the full names of seven particle classes are C1: calcium-rich and soil-dust-like particles; C2: aged-soot-like particles; C3: sodium-salt-like particles; C4: secondary- inorganics-rich and amine containing particles; C5: aged-biomass-burning and soil-dust like particles; C6: aged-biomass-burning and organosulfate-containing particles; C7: mixed/aged and dust-like particles

**Table 2: Features and mean values (±sdev) of characteristic parameters for the two selected episodes**

| | Sodium chloride rich episode | Organics rich episode | Instrument |
|---|---|---|---|
| Date and time | 0820 noon to 0823 noon | 0825 midnight to 0828 midnight | |
| Sodium salts (µg m$^{-3}$) | 2.2 ± 1.0 Higher | 0.6 ± 0.4 Lower | LAAPTOF |
| Biomass burning–soil (µg m$^{-3}$) | 0.7 ± 0.7 Lower | 1.7 ± 1.1 Higher | LAAPTOF |
| Organics (µg m$^{-3}$) | 2.0 ± 1.2 Lowest | 12.9 ± 3.9 Highest | HR-Tof-AMS |
| Particle acidity indicator $NH_4^+$ measured/predicted | 0.67 ± 0.10 Lowest value | 0.83 ± 0.06 Average value Anti-correlated with org | HR-Tof-AMS |
| Organonitrate fraction indicator $NO^+/NO_2^+$ | 11.0 ± 6.1 Highest | 6.1 ± 0.9 Lowest | HR-Tof-AMS |
| Oxidation state (Osc) (Anti-correlated with org) | -0.24 ± 0.34 Less oxidized Strongest variability | -0.10 ± 0.20 More oxidized Least variability | HR-Tof-AMS |
| Wind Direction (°) | Southwest | Varying | Meteorology sensor (WS700, Luft GmbH) |
| Wind Speed (m s$^{-1}$) | 1.6 ± 1.0 | 1.2 ± 1.0 | |

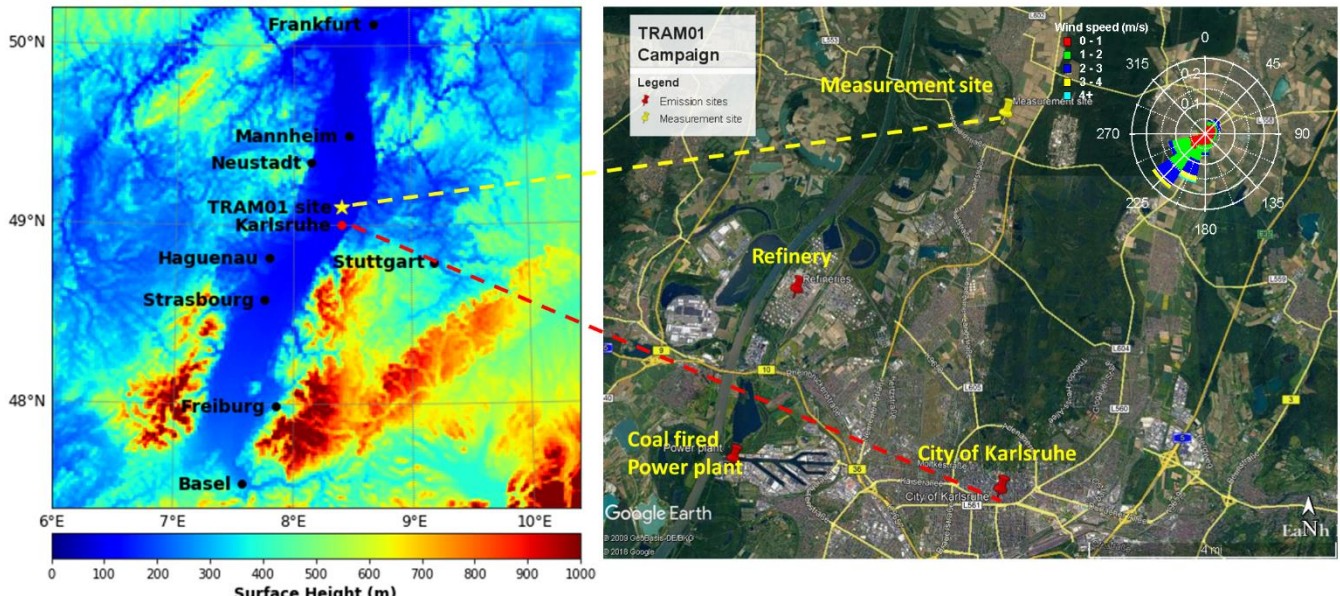

**Figure 1: Measurement location and topography (left), nearby peak emission sites and wind rose for the complete measurement campaign in summer 2016 (right).**

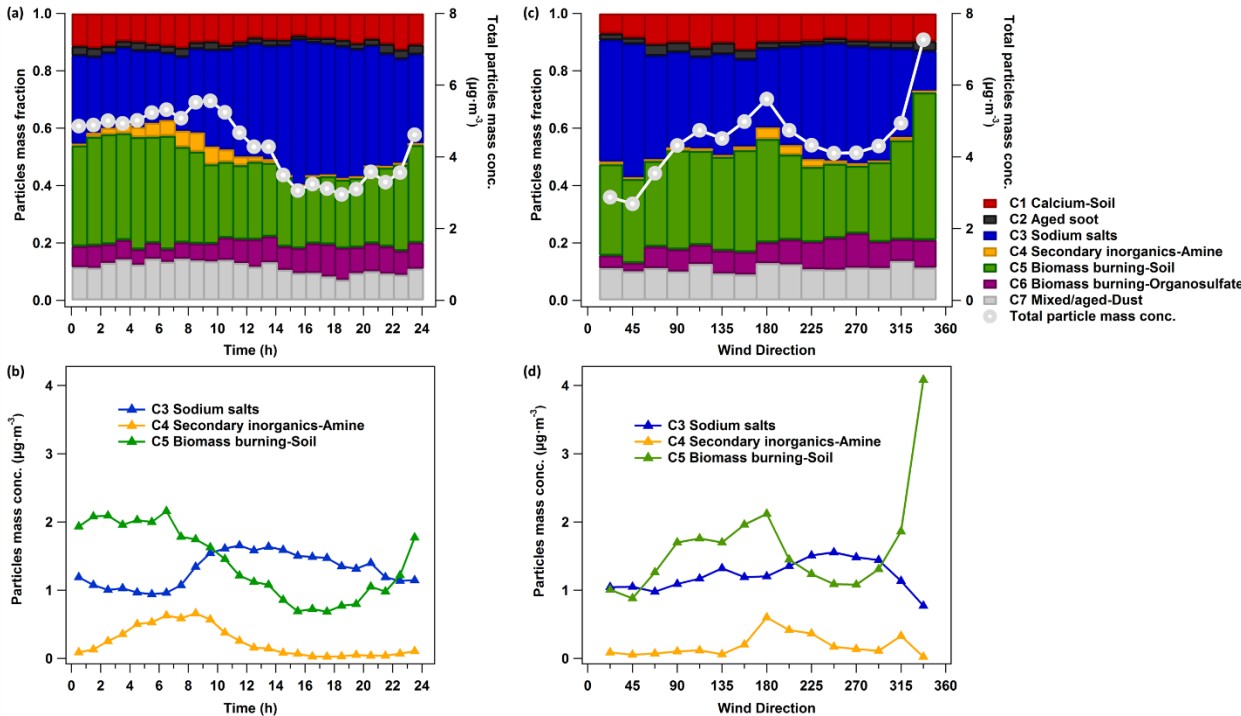

**Figure 2: Diurnal pattern (a and b) and wind direction resolved particle class masses (c and d) measured by LAAPTOF. The data is averaged over the entire campaign. In total 4842 time points of 10 min averaged data were analysed for these plots. The total particle mass concentration for each hour of the day is on average 4.3 ($\pm$ 2.6, one $\sigma$) µg m$^{-3}$ and for each wind direction is on average 4.5 ($\pm$ 2.9) µg m$^{-3}$. For the dominating classes the average mass concentrations are 1.3 ($\pm$ 1.1) µg m-3and 1.2 ($\pm$ 1.0) µg m$^{-3}$ for class 3, 0.2 ($\pm$ 0.5) µg m$^{-3}$ and 0.2 ($\pm$ 0.4) µg m$^{-3}$ for class 4, 1.4 ($\pm$ 1.0) µg m$^{-3}$ and 1.6 ($\pm$ 1.5) µg m$^{-3}$ for class 5, for (b) and (d) respectively. It is worthy to note that particle mass quantification is one of the most challenging issues in single particle mass spectrometry (Shen et al., 2019, and the references therein). There are several assumptions mainly regarding particle shape and density in our particle mass quantification method, leading to potential uncertainties of up to ~300 %. Despite such large uncertainty, the estimated total particle mass has good correlation with the total mass measured by the AMS, covering ~85% of the measurement time (Shen et al., 2019).**

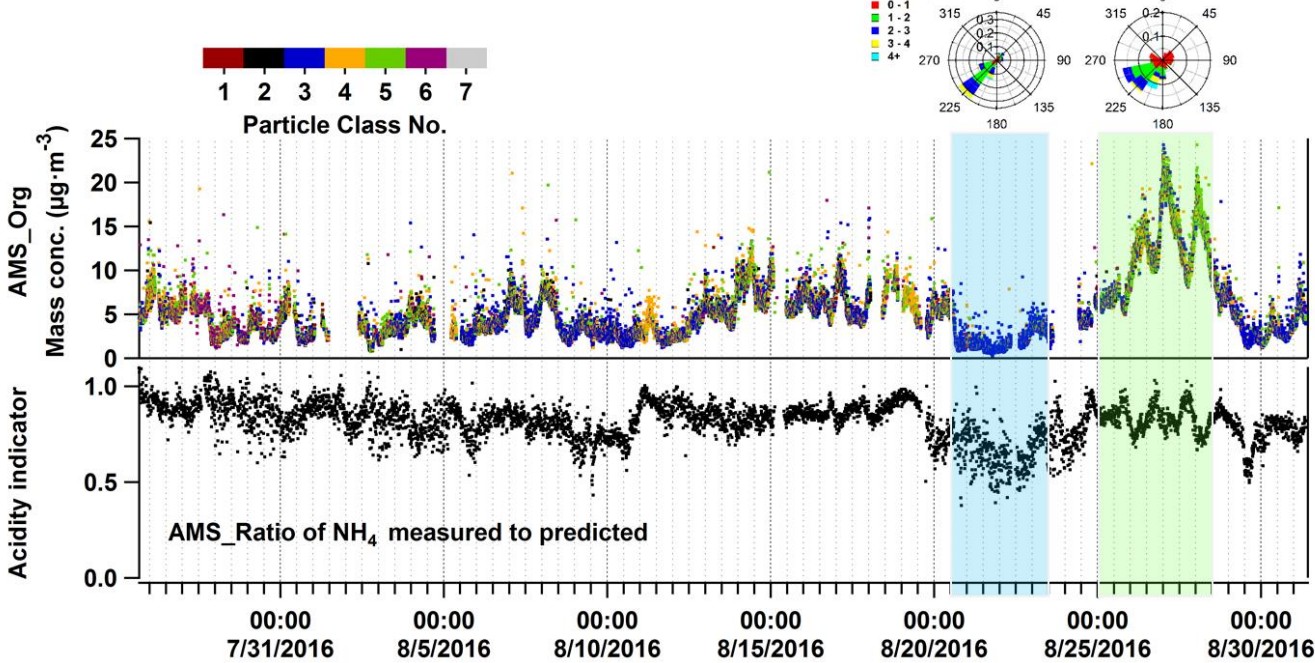

Figure 3: Time series of particulate organic mass measured by the AMS (AMS org in the upper panel), the particle acidity indicator (the lower the value the more acidic, lower panel), and the particle class number classified by LAAPTOF (colour code, upper panel). The seven classes are class 1: calcium-Soil; class 2: aged soot; class3: sodium salts; class 4: secondary inorganics-amine; class 5: biomass burning-soil; class 6: biomass burning-organosulfate; and class 7: mixed/aged-dust. The first (blue) and second (green) shaded areas mark the sodium chloride rich and organic rich episodes, respectively.

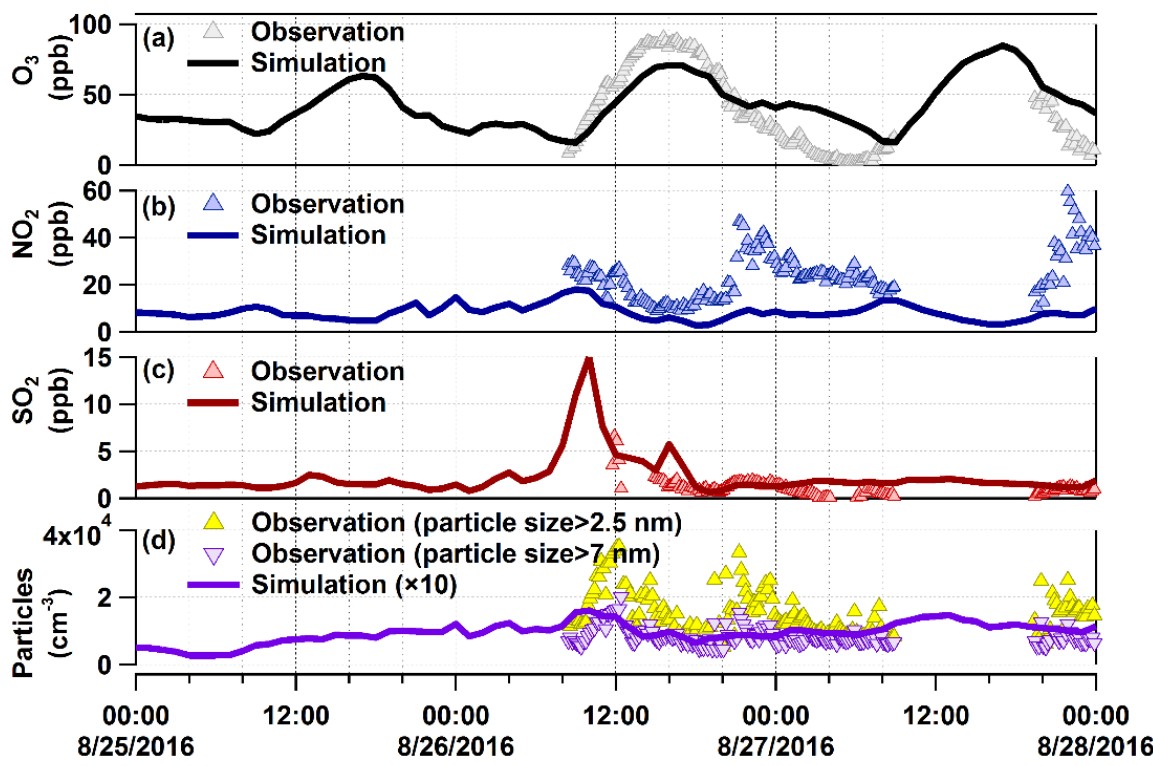

**Figure 4:** Comparison between observations and model results for $O_3$, $NO_2$, $SO_2$, and particle number in organic-rich episode. The modelled particles do not represent the ultrafine particles to their full extent, especially for the particles in the nucleation mode.

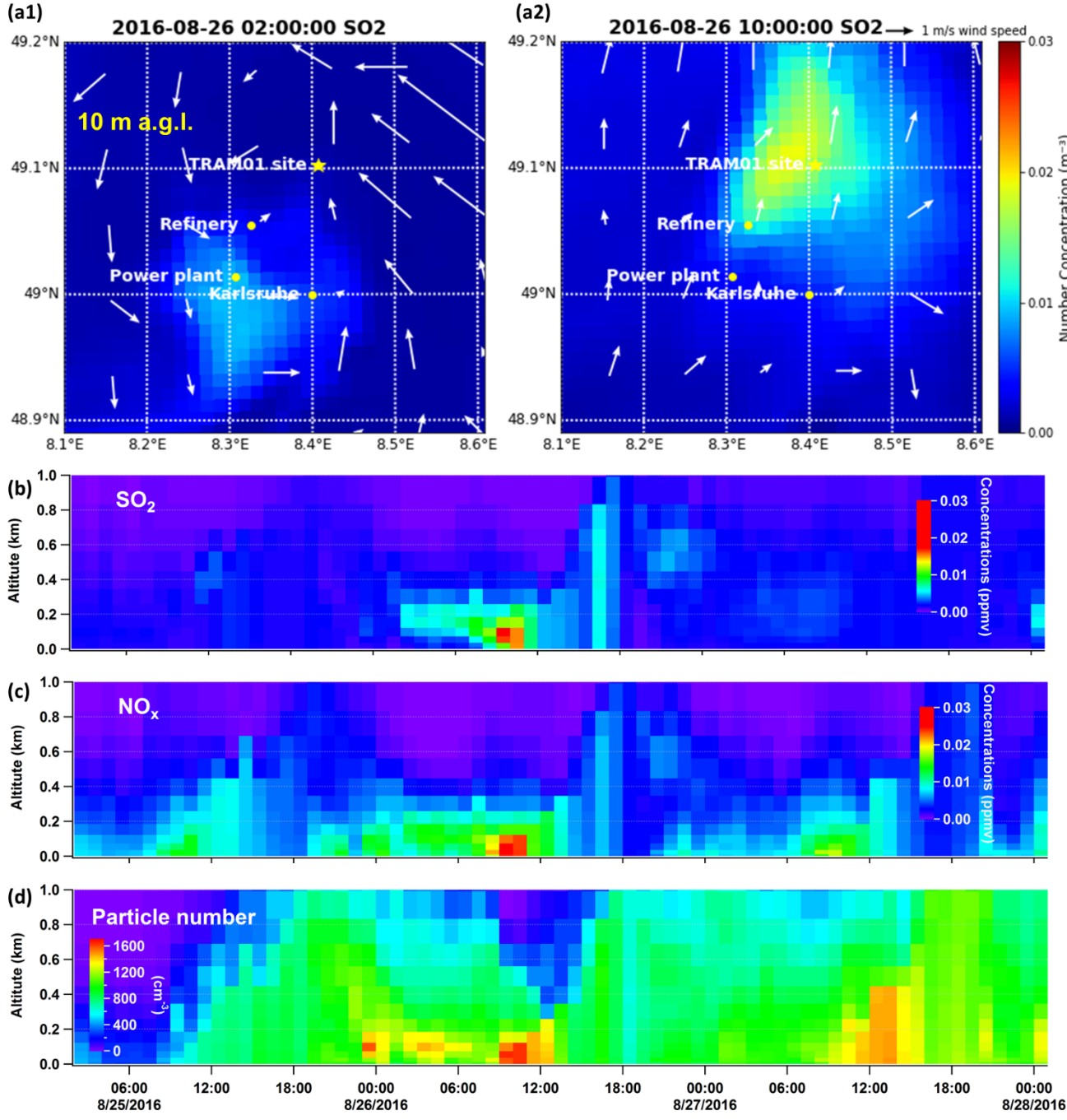

775

**Figure 5: Modelled SO₂ concentrations at 2:00 (a1) and 10:00 (a2) on August 26th at 10 m a.g.l.. Panels (b) to (d) are time series of vertical profiles above the measurement site for simulated SO₂, NOₓ and total particle number.**

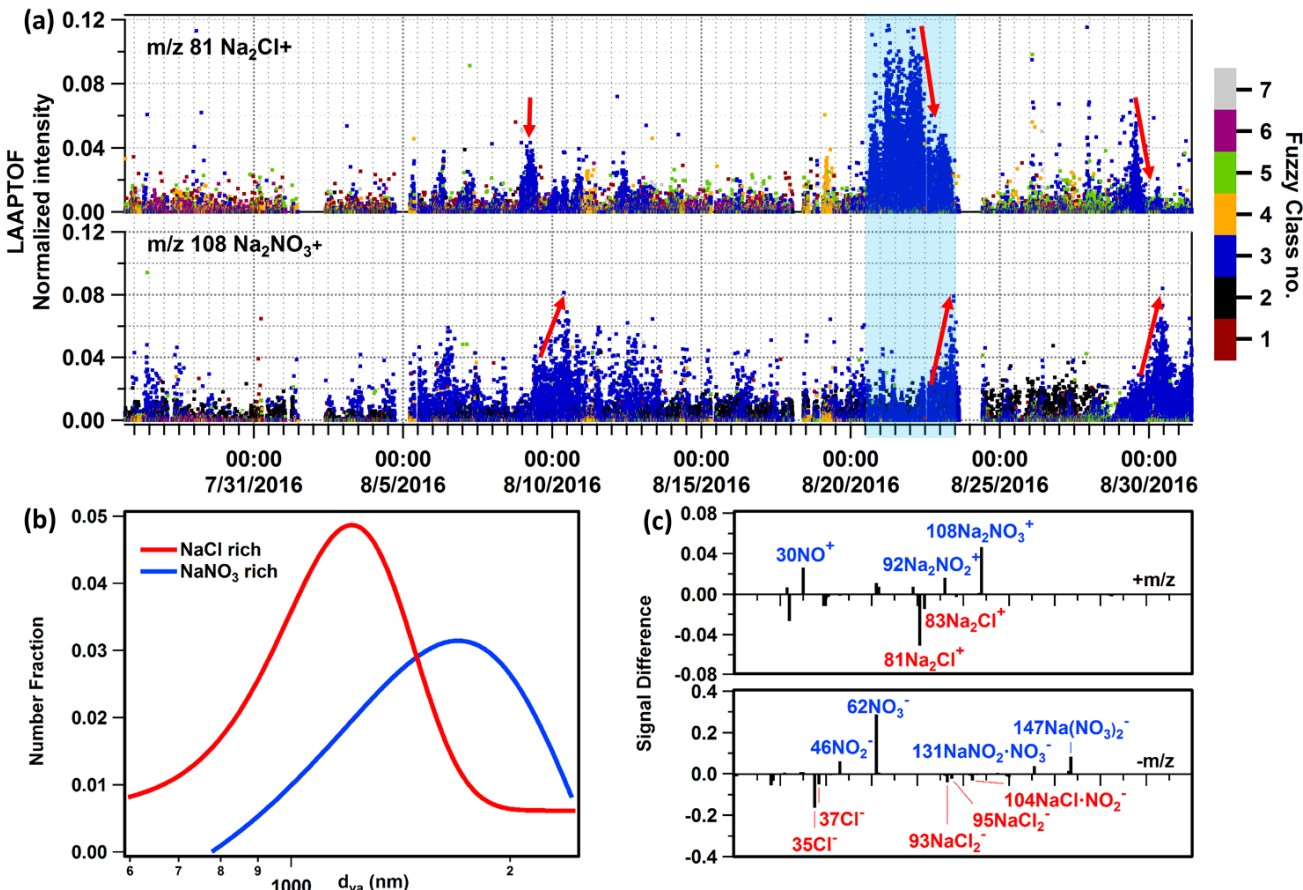

**Figure 6: (a)** Time series of sodium chloride and nitrate signatures in seven particle classes. Y-axis is the normalized ion intensity (each ion peak intensity is normalized to the sum of all ion signals; positive and negative ions were analysed separately). The seven classes are class 1: calcium-Soil; class 2: aged soot; class3: sodium salts; class 4: secondary inorganics-amine; class 5: biomass burning-soil; class 6: biomass burning-organosulfate; and class 7: mixed/aged-dust. Obvious sodium chloride and nitrate signatures are mainly found in class 3 labelled in blue colour. **(b)** Size distributions of NaCl rich (young sea salt) and NaNO₃ rich (aged sea salt) particles. Gaussian fit NaCl $d_{va}$ (±σ) = 1211(±352) nm; NaNO₃ 1697(±946) nm. **(c)** Spectra difference between aged and young sea salt particles (aged-young). The blue band marks the sodium chloride rich episode.

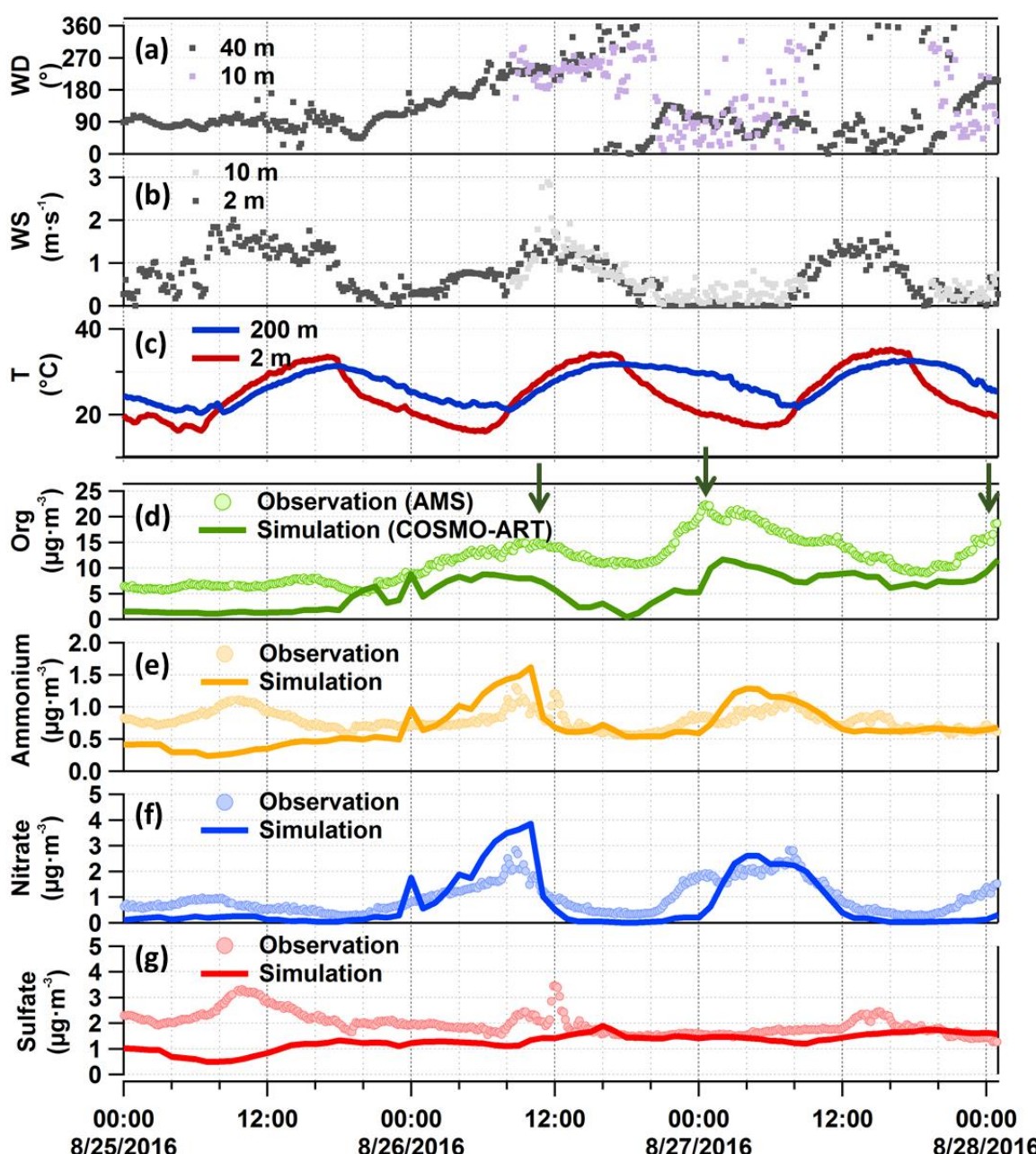

**Figure 7: Comparison between observations and model simulations for organic-rich episode. Panel (a) and (b) are wind conditions measured 10 m a.g.l. at our measurement site (purple and light grey dots for wind direction and speed, respectively), as well as the complementary KIT-tower data (dark grey dots). Panel (c) is the temperature data from the KIT-tower. Panel (d) to (g) are the particulate organic compounds (with green arrows pointing out the high organic peaks), ammonium, nitrate, and sulfate measured by AMS and simulated by COSMO-ART. The simulation results are at the lowest atmospheric level in the model, namely 10 m a.g.l.**

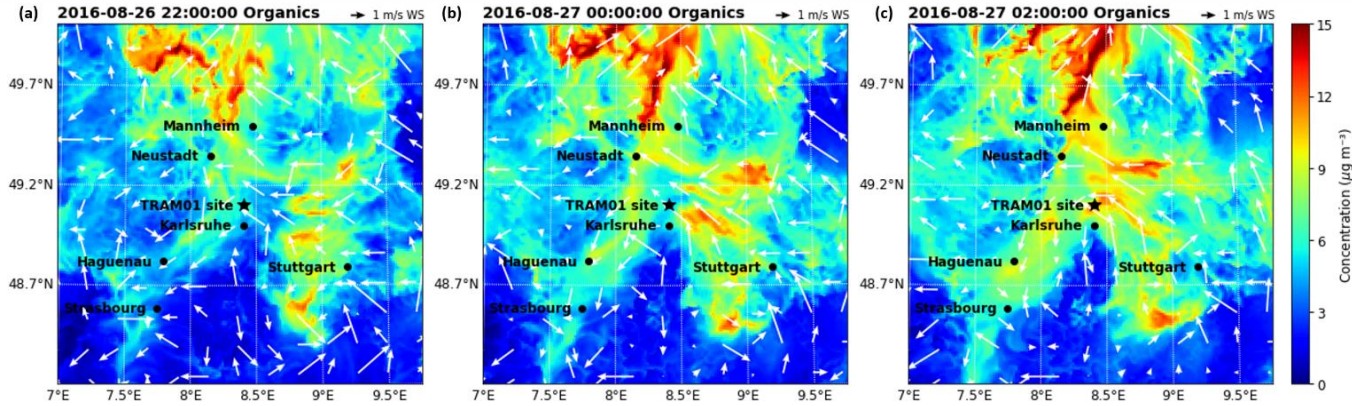

**Figure 8: Particulate organics concentrations calculated by COSMO-ART for three different time points (a) to (c) during the night from August 26th to 27th for 10 m a.g.l..**

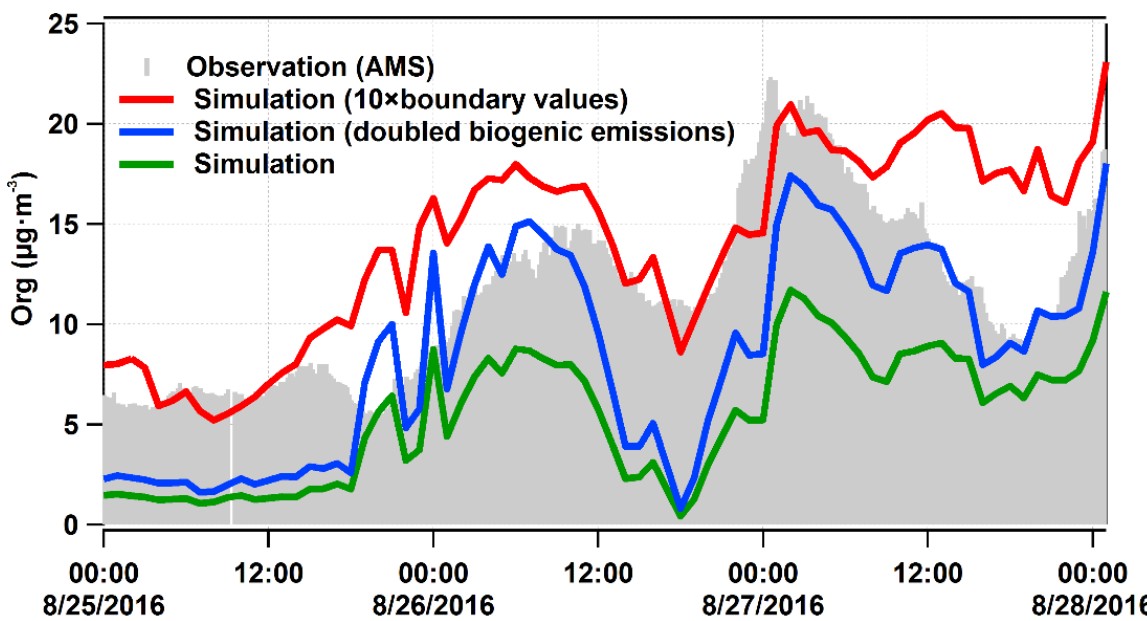

**Figure 9: Modelled organic mass concentrations (lines) compared to measured values (grey area). The green line is the same as in Figure 7, the red line was calculated for a tenfold-enhanced regional background in organics, and the blue was calculated for doubles biogenic emissions in COSMO-ART.**