# Peer review of "Composition and origin of PM2.5 aerosol particles in the upper Rhine valley in summer"

_Atmospheric Chemistry and Physics, 2019_

## Referee Comment (RC1) · Anonymous Referee #1 · 9 Jul 2019

General comments:

The present work conducted a 6 week measurement of particle chemical composition combining LAAPToF and AMS technique at a rural site of southwest Germany. Also the regional transport model COSMO-ART was used to investigate particle sources as well as their transport pattern. The combination of measurement and model can better understand the particle sources and regional transport. The results are straight forward and useful, but some clarification and minor changes are still needed for this manuscript. I suggest a minor revision for this manuscript before publication.

Specific comments: ïĄň Line 18: It's a little confusing with the statement: sodium salts accounted for 36%, but organics, sulfate, nitrate, and ammonium accounted for 58%, 22%, 10%, and 9% (sum up 99%). I believe that the total mass the authors used for

AMS is the sum of org and SNA. While LAAPToF measures not only non-refractory particles. It seems these results cannot be closure, especially for some readers who are not very familiar with these two instruments. Anyway, some clarification is needed to avoid misunderstanding these results. ïĄň Line 123: Do AMS use PM2.5 lenses? ïĄň Line 121: It's no need for this too many references here. ïĄň Line 195: Since the MS discussed long-range transport, I suggest discussing more about secondary aerosols as well. ïĄň Line 188: I suggest to explain a little bit about the particle classes, not only show the names, because not all the readers have read the paper (Shen et al. 2019) ïĄň Session 3.4.1 and 3.4.2: Because SOA is also an important component of fine particles, I suggest discussing more about SOA. Besides, AMS-PMF can provide OOA (SOA). This could be considered in the discussion. ïĄň Session 4 conclusion: More implication should be provided, because this work presents a method of combination of field measurement and model. Can this method be expanded to other places and time? Otherwise this work is only a very local study.

Please also note the supplement to this comment:
https://www.atmos-chem-phys-discuss.net/acp-2019-441/acp-2019-441-RC1-supplement.pdf

---

## Referee Comment (RC2) · Anonymous Referee #2 · 18 Jul 2019

This paper describes a six-week ground-based study of aerosol chemistry in the Rhine Valley, near Karlsruhe. Both LAAPTOF and AMS were deployed and the field data is interpreted in conjunction with COSMO-ART model. This paper is well written, and it is a valuable contribution to ACP. While the paper is somewhat closely related to the author's previous paper, I think that there is enough new material to merit publication. I have some concerns about how AMS data is used, so I recommend some major revisions before publication.

Major comments:

There are some outstanding questions about source apportionment (for example, the biomass burning class identified by LAAPTOF). I think running PMF analysis on AMS data would help answer those questions and make the analysis more complete. There

is certainly enough data to do so and it is an odd choice not to include such analysis.

To add on to the above, a unique feature of this study is that SPMS and AMS are deployed together. It would be nice to see a bit more synergy in data analysis as well. Currently, it seems like the details focus on SPMS. For example, for marine-influenced air masses, did high resolution AMS spectra show typical AMS marine markers, such as MSA?

Minor comments:

Page 4, line 118: "...deployment of a laser ablation aerosol particles time-of-flight mass spectrometer (LAAPTOF; AeroMegt GmbH)" Pretty sure this should be "particle" instead of "particles"

Page 5, line 138: Acronym TSP shows up for the first time and has not been defined.

Page 6, line 180: "behaved anti-correlated" is an odd turn of the phrase. Would revise to "were anti-correlated"

Page 6, line 182: should this read "...for particles larger than 2.5 $\mu$m measured in this study..." instead of 2.5 nm?

For Figure 2 and associated discussion, are these averages over the entire campaign?

Page 8, line 246: averaged -> average

Page 8, line 248: organics -> organic

Figure 4 caption: extend -> extent

In Figure S9, it is hard to tell visually that LAAPTOF m/z 129 and AMS organonitrate are correlated. There are quite a few instances where they are not. Can you plot these against each other and give the R2?

---

## Author Response (AR1)

We gratefully thank the reviewers for their helpful comments to improve the quality of our manuscript. Reviewers' comments are in black. Our point-to-point replies marked by "R" are in blue. Changes to the manuscript text are in green.

**Referee #1 comments:**

General comments:

The present work conducted a 6 week measurement of particle chemical composition combining LAAPToF and AMS technique at a rural site of southwest Germany. Also the regional transport model COSMO-ART was used to investigate particle sources as well as their transport pattern. The combination of measurement and model can better understand the particle sources and regional transport. The results are straightforward and useful, but some clarification and minor changes are still needed for this manuscript. I suggest a minor revision for this manuscript before publication.

Specific comments:

1. Line 18: It's a little confusing with the statement: sodium salts accounted for 36%, but organics, sulfate, nitrate, and ammonium accounted for 58%, 22%, 10%, and 9% (sum up 99%). I believe that the total mass the authors used for AMS is the sum of org and SNA. While LAAPToF measures not only non-refractory particles. It seems these results cannot be closure, especially for some readers who are not very familiar with these two instruments. Anyway, some clarification is needed to avoid misunderstanding these results.

R: Yes, the total mass used for AMS is the sum of organics and SNA, as well as chloride. In contrast, the total mass determined by LAAPTOF is the sum of 7 particle types. We have revised this sentence for clarification, as follows:

"For the entire measurement period, the total aerosol particle mass was dominated by sodium salts contributing on average (36 ± 27) % to the total single particle mass measured by the LAAPTOF. The total particulate organic compounds, sulfate, nitrate, and ammonium contributed on average (58 ± 12) %, (22 ± 7) %, (10 ± 1) %, and (9 ± 3) % to the total non-refractory particle mass measured by the AMS."

2. Line 123: Do AMS use PM2.5 lenses?

R: AMS and LAAPTOF are both are equipped with PM2.5 aerodynamic inlet lenses.

3. Line 121: It's no need for this too many references here.

R: We have removed some references here. The revised sentence is as follows:

"The LAAPTOF is a commercially available single particle mass spectrometer and has been well described in recent publications (Gemayel et al., 2016; Marsden et al., 2016; Ramisetty et al., 2018; Shen et al., 2018)."

4. Line195: Since the MS discussed long-range transport, I suggest discussing more about secondary aerosols as well.

R: The long-range transport we discuss is related to a unique episode where the sodium salts accounted for (70 ± 24) % of the total mass of the single particles measured by the LAAPTOF. In addition, the mass concentrations of organics and secondary inorganics (ammonium, sulfate, and nitrate) measured by the AMS were at the lowest level compared to the other periods during the entire measurement campaign. During the long-range transport

episode secondary species mainly formed on the sodium salt particles e.g. $NaNO_3$. This is discussed regarding sub-classes and the sea salt aging process in section 3.3 "Aged sea salt in central Europe".

5. Line 188: I suggest to explain a little bit about the particle classes, not only show the names, because not all the readers have read the paper (Shen et al. 2019)

R: We have added more explanations, as follows:

"In a previous publication (Shen et al., 2019), we could identify seven major particle classes at the measurement site among ~$3.7 \times 10^5$ single particles with sizes between 200 nm and 2.5 μm $d_{va}$. In brief, Class 1 are calcium-rich and soil-dust-like particles (calcium-soil) with marker peaks at m/z 40 $Ca^+$, 56 $CaO^+$, 57 $CaOH^+$, 75 $CaCl^+$, 96 $Ca_2O^+$, and 112 $(CaO)_2^+$. This particle class shows a good correlation (Pearson's correlation coefficient $\gamma \geq 0.6$) with the laboratory derived reference spectra of soil dust particles. Class 2 are aged-soot-like particles (aged soot) with characteristic $C_n^\pm$ patterns and are mixed with sulfate and nitrate. Class 3 are sodium-salt-like particles (sodium salts) with marker peaks at m/z 23 $Na^+$, 81/83 $Na_2Cl^+$, 139/141 $Na_3Cl_2^+$, 165 $Na_3SO_4^+$, 35/37 $Cl^-$, 93/95 $NaCl_2^-$, 115 $Na(NO_2)_2^-$, 131 $NaNO_2NO_3^-$, and 147 $Na(NO_3)_2^-$. Class 4 are "Secondary inorganic–amine" particles with prominent ammonium, nitrate and sulfate markers, as well as amine marker peaks at m/z 58 $C_2H_5NHCH_2^+$, 59 $(CH_3)_3N^+$, 86 $(C_2H_5)_2NCH_2^+$, 88 $(C_2H_5)_2NO/C_3H_6NO_2^+$, and 118 $(C_2H_5)_2NCH_2^+$. Class 5, aged-biomass-burning and soil-dust like particles (biomass burning–soil), has aged biomass burning marker peaks at m/z 39 $K^+$, 213 $K_3SO_4^+$, 46 $NO_2^-$, 62 $NO_3^-$, 97 $HSO_4^-$, and peaks related to aromatic compounds (50 $C_4H_2^+$, 63 $C_5H_3^+$, 77 $C_6H_5^+$, 85 $C_7H^+$, 91 $C_7H_7^+$, 95 $C_7H_{11}^+$, 104 $C_8H_8^+$, 115 $C_9H_7^+$). Furthermore, class 5 shows a strong correlation with reference spectra of soil dust particles. The representative spectra of class 6, aged-biomass-burning and organosulfate-containing particles (biomass burning–organosulfate), also features aged biomass burning particles and has organosulfates marker peaks at m/z 141 $C_2H_5OSO_4^-$, 155 $C_2H_3O_2SO_4^-$, and 215 $C_5H_{11}OSO_4^-$. Class 7 particles contain all the marker peaks from the other six classes and show a good correlation with the reference spectra of mineral and soil dust particles, and are consequently labelled as "mixed/aged dust" (short for mixed/aged and dust-like particles) …"

6. Session 3.4.1 and 3.4.2: Because SOA is also an important component of fine particles, I suggest discussing more about SOA. Besides, AMS-PMF can provide OOA (SOA). This could be considered in the discussion.

R: Thank you for your suggestion. We have done a PMF analysis, added more explanation in the introduction, added one subsection in method part, added more discussion about SOA in section 3.1 and 3.4, put the corresponding results in the supplement, and modified the conclusions accordingly. The corresponding changes are listed as follows:

**Abstract**

"…Positive matrix factorization (PMF) analysis for the AMS data suggests that the total organic aerosol (OA) consisted of five components, including (9 ± 7) % hydrocarbon-like OA (HOA), (16 ± 11) % semi-volatile oxygenated OA (SV-OOA), and (75 ± 15) % low-volatility oxygenated OA (LV-OOA) …"

**Section 1 Introduction (3rd paragraph)**

"…According to PMF analysis, the organic aerosol (OA) can be separated into e.g., hydrocarbon-like OA (HOA), cooking-related OA (COA), nitrogen-enriched OA (NOA), biomass burning OA (BBOA), and semi-volatile

oxygenated OA (SV-OOA), and low-volatility oxygenated OA (LV-OOA). This information hints to OA aging and potential sources…"

**Section 2.2 Mass spectrometer data analysis**

"Single particle data were analysed by using the Igor LAAPTOF data analysis software (version 1.0.2, AeroMegt GmbH). The corresponding basic procedures (e.g., mass calibration and fuzzy c-means clustering) and particle mass quantification method have been described in detail in our previous studies (Shen et al., 2018 and 2019). The AMS data analysis was done with the standard Igor software, including SQUIRREL (version 1.60C) and PIKA (version 1.20C). To account for a particle bouncing effect, we applied a composition-dependent collection efficiency (CE, the product of net particle transmission and detection efficiency) to the entire AMS dataset (Canagaratna et al., 2007; Middlebrook et al., 2012). In order to investigate the potential sources for organic aerosols, we have done positive matrix factorization (PMF) analysis for AMS high-resolution (HR) spectra of organic compounds, by using the PMF Evaluation Tool (PET version 3.00D) (Ulbrich et al., 2009). Detailed protocols were reported by Ulbrich et al. (2009) and (DeCarlo et al., 2010). In brief, the HR input matrices without isotopes were created using PIKA. A five-factor-PMF solution with FPEAK=0 was chosen according to the following criteria, including least residuals (unexplained data), $Q/Q_{exp} \approx 1$ (Q is the sum of the squares of the scaled residuals; $Q_{exp}$ is the expected Q), and distinguishable features (e.g., spectra and diurnal patterns) of the components. It should be noted again that although fuzzy clustering and PMF analyses are commonly used methods in source apportionment, they cannot provide the information of aerosols origin and the interaction between aerosol transport and transformation processes, which can be studied with aerosol transport models."

**Section 3.1 Overview of the field observation (2$^{nd}$ and last paragraph)**

"…Given the AMS-PMF results shown in Fig. S4, five OA factors (components) were determined with characteristic mass spectra and corresponding elemental ratios (O:C and H:C), and distinct temporal variations. There were (9 ± 7) % HOA, (16 ± 11) % SV-OOA, and (75 ± 15) % LV-OOA, i.e., (16 ± 15) % LV-OOA1, (37 ± 17)% LV-OOA2, and (21 ± 13) % LV-OOA3, contributing to the OA mass. The HOA and OOA are linked to primary and secondary organic aerosol (POA and SOA), respectively (Ulbrich et al., 2009). Thus, the SOA dominated the OA mass at this measurement site. Further descriptions of PMF results can be found in the supplement (Fig. S4) and more discussions are given in section 3.4.2 especially regarding the organic compounds. …

… Furthermore, we discuss another episode with relatively high organic mass concentrations to find out how these mass loads could form and exist for a relatively long time period. Both episodes are easily recognized from the time series of particles measured by LAAPTOF and the organic compounds measured by AMS (Fig. S2 and Fig. 3; see also Fig. 4 in Shen et al, 2019), as well as the time series of OOA, in particular for SV-OOA, LV-OOA1 and LV-OOA2 (Fig. S4) …"

**Section 3.4.2 Organic compounds (1$^{st}$ paragraph)**

"The dominating OA components in this episode were LV-OOA1 (42 ± 15) % and LV-OOA2 (32 ± 21) %, which

are surrogates for more aged SOA (Zhang et al., 2011). It should be noted that LV-OOA1 has strong correlation with SV-OOA ($\gamma = 0.8$), which is a surrogate for local (less-aged) SOA (Zhang et al., 2011). As shown in Fig. S4 (the insert in panel d), they are both peaking at a similar time during the night with broader peaks for LV-OOA1. This indicates that the LV-OOA1 might have a similar origin as the SV-OOA (more related with local biogenic sources). However, it can also be influenced by regional transport. For example, the first LV-OOA1 peak has two sub-peaks: the first one appears during the night, e.g., related to $NO_3$ oxidation (strong correlation with $\gamma = 0.9$ between LV-OOA1 and CHO1N family has been found here, refer to their time series in Fig. S4 b); and the second sub-peak appears in the morning rush hour and corresponds to south and southwest wind directions, indicating the potential transport from the urban and industrial area of Karlsruhe. The more oxidized LV-OOA2 is peaking during day-time and shows a negative correlation with SV-OOA ($\gamma = -0.5$), possibly due to photochemical aging of SV-OOA. Therefore, we conclude that the SOA measured in this study was likely due to both local formation and regional transport."

**Section 4 Conclusion (2nd and 4th paragraphs)**

"…AMS-PMF results suggest the organic aerosols (OA) consisted of $(9 \pm 7)$ % HOA, $(16 \pm 11)$ % SV-OOA, $(75 \pm 15)$ % LV-OOA with three sub-types (LV-OOA1, LV-OOA2, and LV-OOA3).

…

For the organics rich episode, the particulate organic compounds accounted for $(77 \pm 6)$ % of total non-refractory particle mass measured by the AMS. AMS-PMF analysis suggests the dominating OA components were LV-OOA1 and LV-OOA2, contributing $(42 \pm 15)$ % and $(32 \pm 21)$ % to the total OA mass in this episode. LV-OOA1 and LV-OOA2 had positive ($\gamma = 0.9$) and negative ($\gamma = -0.5$) correlation with SV-OOA, respectively, indicating their local formation. In order to investigate origin of the aerosols and their transport and transformation processes, we applied a regional transport model, COSMO-ART. …"

**Supplement**

[Figure]

**Figure S4: Five-factor-PMF solution with FPEAK=0 for organic compounds measured by AMS during the whole measurement time. (a) Mass spectra of the five components, i.e., HOA, SV-OOA, LV-OOA1, LV-OOA2, and LV-OOA3. (b) Time series of the five components mass concentrations (left Y-axis) and some related species (right Y-axis). (c) Diurnal patterns of the five components. (d) Time series of the components with the same data as panel but stacked (b). The percentages are their contributions to OA. Note that the total OA value showed here is lower than the input OA for the PMF analysis. This is because the components identification might be interfered by extraneous variability, arising from some causes such as instrumental issues. Such extraneous variability will lead to disproportionate effects on the fitting outcome. Therefore, we need to downweight the corresponding variables in a proper way or even remove them, resulting the lower output total OA value reconstructed by the components.**

HOA is associated with POA, e.g., from urban emissions. In this study, the HOA (high H:C = 1.84 and low O:C = 0.12, herein) has a similar trend as m/z 57 (major contribution from $C_4H_9$) and $NO_2$ (not shown), with Pearson's correlation coefficients γ = 0.5 and 0.4, respectively. SV-OOA (O:C = 0.32) shows a very strong correlation with m/z 91 (mainly from $C_7H_7$) (γ = 0.9) and CH family (γ = 0.8). Three LV-OOA have prominent marker peaks at m/z 28 and 44, and all have high O:C ratio (> 0.7), which indicates highly oxidized or aged OA. LV-OOA1 shows strong correlations (γ = 0.8 to 1) with CH, CHO1, CHOgt1, CHN, CHO1N, and CHOgt1N. LV-OOA2 shows good correlations with CHOgt1N (γ = 0.6). LV-OOA3 shows no good correlation with any ion families, but a similar trend as sulfate (γ = 0.5).

Similar elemental ratios for HOA, SV-OOA, and LV-OOA and their relationships with tracers such as HOA with NOx and LV-OOA with secondary inorganics, have also been reported in previous AMS-PMF studies. Readers may refer to a review paper by Zhang et al. (2011).

7. Session 4 conclusion: More implication should be provided, because this work presents a method of combination of field measurement and model. Can this method be expanded to other places and time? Otherwise this work is only a very local study.

R: Yes, this combination method can be expanded to other places and time. We have added more implications in ours conclusion, as follows:

**2nd last paragraph**

"…Such field-model comparisons and corresponding sensitivity tests are useful for targeting missing sources and mechanisms in the current model and thus help its development. Such that model can be improved to become better predictive, which is useful for e.g., legislation and implementation of policies against air pollution. Consequently, model systems suitable for a reasonable environmental policy should be validated by comprehensive field studies with state-of-the-art aerosol analytics and meteorological parameters."

**Last paragraph**

"…Our results show how close the physical and chemical nature of aerosols is related to emission sources, transformation processes, and complex transport pattern. The method of combination of filed measurement and model calculation in this study can be widely used in other places in the world, in order to investigate the origin of aerosols and their interaction between transport and transformation. For example, in some pollution hot spots like Po Valley, and some other places which are subject to both biogenic and anthropogenic emissions. Besides, it is also not limited to the summer time that we studied…"

**Reply to Referees**

**Referee #2 comments:**

This paper describes a six-week ground-based study of aerosol chemistry in the Rhine Valley, near Karlsruhe. Both LAAPTOF and AMS were deployed and the field data is interpreted in conjunction with COSMO-ART model. This paper is well written, and it is a valuable contribution to ACP. While the paper is somewhat closely related to the author's previous paper, I think that there is enough new material to merit publication. I have some concerns about how AMS data is used, so I recommend some major revisions before publication.

**Major comments:**

1. There are some outstanding questions about source apportionment (for example, the biomass burning class identified by LAAPTOF). I think running PMF analysis on AMS data would help answer those questions and make the analysis more complete. There is certainly enough data to do so and it is an odd choice not to include such analysis.

R: Thank you for your suggestion. We have done PMF analysis for organics data measured by the AMS, added more explanation and discussion throughout the whole manuscript, and modified the conclusion as well. Please refer to our reply to the 1st Referee comment #6.

In addition, we couldn't identify biomass burning organic aerosols (BBOA) for the entire measurement period, based on AMS-PMF results. We have added this point in the last second paragraph of section 3.1, as follows:

"… It should be noted that, there were no significant biomass burning organic aerosols (BBOA) identified by PMF for the entire measurement period. The marker f60 (the mass fraction of m/z 60 to the total organics) of BBOA being very close to the background fraction of 0.003 (Crippa et al., 2014; Cubison et al., 2011). In addition to biomass burning, also other potential sources for potassium and sulfate rich particles detected by LAAPTOF must be taken into consideration, such as coal combustion (Xu et al., 2018). Therefore, further analysis is required to identify the different sources contributing to this particle class."

2. To add on to the above, a unique feature of this study is that SPMS and AMS are deployed together. It would be nice to see a bit more synergy in data analysis as well. Currently, it seems like the details focus on SPMS. For example, for marine-influenced air masses, did high resolution AMS spectra show typical AMS marine markers, such as MSA?

R: Thank you for your suggestion. We have added more synergy in data analysis regarding the marine influenced air masses. In high resolution AMS spectra we have seen the peak at 78.99 $CH_3SO_2+$, which is reported to be the marker for methanesulfonic acid (MSA) (Huang et al., 2015). Based on the method developed by Huang et al. (2015), we have also estimated the MSA fraction and mass concentrations for the entire campaign and compare them with sodium salts measured by the LAAPTOF. We have added the corresponding discussion and figure are as follows:

**Section 3.3 Aged sea salt in central Europe (last paragraph)**

"Apart from that, in high resolution AMS spectra we have observed the peak at 78.99 $CH_3SO_2^+$, which is the marker for methanesulfonic acid (MSA) from marine or coastal areas (Huang et al., 2015) although other sources cannot be excluded (Daellenbach et al., 2017). Based on the method developed by Huang et al. (2015), we have also estimated the MSA fraction ($f_{MSA}$, signal fraction of MSA to total org and sulfate) and mass concentrations

for the entire campaign (refer to Fig. S12). It turns out that during the sodium chloride rich episode, the $f_{MSA}$ is enhanced compared to the other low sodium salts time period, such as organic rich episode. This also supports that sodium salts particles measured by the LAAPTOF had sea salt origin."

**Supplement**

[Figure]

**Figure S12: Time series of the estimated mass concentration of sodium salts particles measured by the LAAPTOF (a), methanesulfonic acid (MSA) fraction to total organics and sulfate measured by the AMS (b) and MSA mass concentration (c). The first (blue) and second (green) shaded areas mark the sodium chloride rich and organic rich episodes, respectively.**

The method used to estimate MSA fraction ($f_{MSA}$, signal fraction of MSA to total org and sulfate) and its mass concentration can be referred to Huang et al. (2015). In brief, W-mode $f_{MSA}$ is calculated and then used to multiply V-mode mass concentration of organics and sulfate, resulting estimated MSA mass concentration. The corresponding equations are as follows:

$$\sum I_{MSA} = \frac{I_{CH3SO2}}{9.7\%} \tag{S2}$$

$$f_{MSA} = \frac{\frac{\sum I_{MSA}}{RIE_{MSA}}}{\frac{\sum I_{Org}}{RIE_{Org}} + \frac{\sum I_{SO4}}{RIE_{SO4}}} \tag{S3}$$

$$C_{MSA} = f_{MSA} \times (C_{Org} + C_{SO4}) \tag{S4}$$

where $\sum I_{MSA}$ is the total signal intensity of all MSA fragments measured in W-mode, which can be estimated based on the fragmentation patterns determined from the laboratory experiments (Huang et al., 2015). It was reported that the signal intensity of marker peak at m/z 78.99 ($CH_3SO_2^+$) accounted for $(9.7 \pm 1.6)$ % of the total signal intensity of all MSA fragments. $\sum I_{Org}$ and $\sum I_{SO4}$ are total signal intensity of organics and sulfate fragments measured in W-mode, respectively. Relative ion efficiency of MSA ($RIE_{MSA}$=1.3) is the averaged value from $RIE_{Org}$ (=1.4) and $RIE_{SO4}$ (=1.2). $C_{Org}$ and $C_{SO4}$ are mass concentrations calculated from V-mode data. $C_{MSA}$ is the estimated MSA mass concentration.

**Reply to Referees**

**Minor comments:**

3. Page 4, line 118: ": : :deployment of a laser ablation aerosol particles time-of-flight mass spectrometer (LAAPTOF; AeroMegt GmbH)" Pretty sure this should be "particle" instead of "particles"

R: Yes, we have revised it.

4. Page 5, line 138: Acronym TSP shows up for the first time and has not been defined.

R: We have replace Acronym TSP with the full name "total suspended particulates"

5. Page 6, line 180: "behaved anti-correlated" is an odd turn of the phrase. Would revise to "were anti-correlated"

R: We have revised it to "were anti-correlated"

6. Page 6, line 182: should this read ": : :for particles larger than 2.5 μm measured in this study: : :" instead of 2.5 nm?

R: This particle number concentration was measured by CPC3776 (TSI Inc.) which has the cut off at 2.5 nm rather than 2.5 μm. We have added this point to make it clear, as follows:

"The averaged particle number concentration of $(1.2 \pm 0.8) \times 10^4 \, cm^{-3}$ for particles larger than 2.5 nm measured by CPC3776 in this study"

7. For Figure 2 and associated discussion, are these averages over the entire campaign?

R: Yes. We have added "the data is averaged over the entire campaign" in the figure caption and discussion part (the third paragraph of section 3.1).

8. Page 8, line 246: averaged -> average

R: We have revised it.

9. Page 8, line 248: organics -> organic

R: We have revised it.

10. Figure 4 caption: extend -> extent

R: We have revised it

11. In Figure S9, it is hard to tell visually that LAAPTOF m/z 129 and AMS organonitrate are correlated. There are quite a few instances where they are not. Can you plot these against each other and give the R2?

R: We agree that it is hard to tell they are correlated. Due to complex laser desorption and ionization (LDI) mechanisms, the resulting ion fragments usually do not have good correlation or direct clear relationship with the results from quantitative instruments such as AMS. This is also the case for LAAPTOF m/z 129 and AMS organonitrate. Specific relationships of LAAPTOF ion intensities and AMS mass concentrations for non-refractory compounds were found based on specific calculation, such as the fraction of org/(org+nitrate), reported by our previous study (Shen et al., 2019). However, in this study, we aimed to confirm that the m/z 129 $C_5H_7NO_3^+$ signal is mainly due to organonitrates rather than the recombination of nitrate and organic fragments due to the LDI matrix effect in single-particle MS. First, we found that during sodium chloride rich episode, organic mass

concentration measured by the AMS was relatively low. However, at the end of this episode m/z 129 $C_5H_7NO_3+$ relative intensity was obviously strong, which was comparable with that observed in high organic episode. Furthermore, we referred to AMS results as well as CIMS results (Huang et al., 2019), both of which show high organonitrate mass concentration at the similar time period as LAAPTOF_ m/z 129 shows high relative intensity (as shown in the figured below). Therefore, we can conclude that m/z 129 $C_5H_7NO_3+$ was mainly contributed by organonitrate.

[Figure]

We have added more explanation in the last paragraph of section 3.3, as follows:

[revised manuscript text omitted]

**Figure S4: Five-factor-PMF solution with FPEAK=0 for organic compounds measured by AMS during the whole measurement time. (a) Mass spectra of the five components, i.e., HOA, SV-OOA, LV-OOA1, LV-OOA2, and LV-OOA3. Peaks are coloured with respect to different ion families. (b) Time series of the five components mass concentrations (left Y-axis) and their tracers or related ion family (right Y-axis). (c) Diurnal patterns of the five components. (d) Time series of the stacked components with the same data as panel (b), but the contribution of the five components can be visualized clearly. The percentages are their contributions to OA. Note that the total OA value showed here is lower than the input OA for the PMF analysis. This is because the components identification might be interfered by extraneous variability, arising from some causes such as instrumental issues. Such extraneous variability will lead to disproportionate effects on the fitting outcome. Therefore, we need to downweight the corresponding variables in a proper way or even remove them, resulting the lower output total OA value reconstructed by the components.**

HOA is associated with POA, e.g., from urban emissions. In this study, the HOA (high H:C = 1.84 and low O:C = 0.12, herein) has a similar trend as m/z 57 (major contribution from $C_4H_9$) and $NO_2$ (not shown), with Pearson's correlation coefficients $\gamma$ = 0.5 and 0.4, respectively. SV-OOA (O:C = 0.32) shows a very strong correlation with m/z 91 (mainly from $C_7H_7$) ($\gamma$ = 0.9) and CH family ($\gamma$ = 0.8). Three LV-OOA have prominent marker peaks at m/z 28 and 44, and all have high O:C ratio (> 0.7), which indicates highly oxidized or aged OA. LV-OOA1 shows strong correlations ($\gamma$ = 0.8 to 1) with CH, CHO1, CHOgt1, CHN, CHO1N, and CHOgt1N. LV-OOA2 shows good correlations with CHOgt1N ($\gamma$ = 0.6). LV-OOA3 shows no good correlation with any ion families, but a similar trend as sulfate ($\gamma$ = 0.5).

Similar elemental ratios for HOA, SV-OOA, and LV-OOA and their relationships with tracers such as HOA with NOx and LV-OOA with secondary inorganics, have also been reported in previous AMS-PMF studies. Readers may refer to a review paper by Zhang et al. (2011).

[Figure]

**Figure S5:** Comparison of particle classes measured by LAAPTOF as a function of wind direction for weekdays (left) and weekends (right). The data is averaged over the entire campaign.

[Figure]

**Figure** S6**: (a) Time series of aged-biomass-burning and soil-dust like particles and black carbon (black and red dots are weekend data); red dots are in organics rich episode. The shaded area marks the organics rich episode. (b) Comparison of mass concentration between biomass burning-soil and black carbon measured during the weekends. Black and red dots are the data without and with organic rich episode, respectively. (c) Diurnal pattern of particle mass concentration of biomass burning-soil (BS) and black carbon (BC) during organics rich episode, respectively.**

[Figure]

**Figure S7:** Time series of organonitrate indicator (grey dots) and oxidation state (black dots). The first (blue) and second (green) shaded areas mark the sodium chloride rich and organic rich episodes, respectively.

[Figure]

**Figure S8: Hybrid Single-Particle Lagrangian Integrated Trajectory (HYSPLIT) back-trajectory analysis for 72 hours for sodium chloride rich (a) and organics rich (b) episodes, respectively.**

**Figure S9: Spectra of the sodium salts class (Shen et al., 2019) and its sub-classes: NaCl rich (averaged from 1487 particles) and NaNO₃ rich (averaged from 300 particles).**

[Figure]

**Figure S9S10: Time series of sodium chloride, nitrate, and organonitrate. Y-axis is the normalized ion intensity (each ion peak intensity is normalized to the sum of all ion signals; positive and negative ions were analysed separately). The seven classes are class 1: calcium-Soil; class 2: aged soot; class3: sodium salts; class 4: secondary inorganics-amine; class 5: biomass burning-soil; class 6: biomass burning-organosulfate; and class 7: mixed/aged-dust. Obvious sodium chloride and nitrate signatures are mainly found in class 3 labelled in blue colour. The first (blue) and second (green) bands mark the sodium chloride rich and organic rich episodes, respectively.**

The method to estimate organonitrate based on AMS measurement can be referred to Farmer et al. (2010) and the parameters used here are the same as that used by Huang et al. (2019). The result in Fig. S8 is based on our AMS measurements and the corresponding equation suggested by Farmer et al. (2010), as follows:

$$y=[(R_{obs}-R_{an})(1+R_{on})]/[(R_{on}-R_{an})(1+R_{obs}]],)] \tag{S1}$$

where y is the estimated organonitrate mass concentration (µg m⁻³), $R_{obs}$ is the observed ambient ratio of $NO_2^+/NO^+$; $R_{an}$ and $R_{on}$ are the $NO_2^+/NO^+$ ratio for ammonium nitrate and organonitrate, respectively. In our study $R_{an}$=0.43 based on the measurement and $R_{on}$=0.1 based on literature data. More details can be found by Huang et al. (2019).

[Figure]

**Figure S11: Laboratory evidence for anthropogenic organonitrate (LAAPTOF signature m/z 129 $C_5H_7NO_3^+$). (a) to (f) are the averaged spectra from several thousand single particles, respectively. Y-axis is the normalized ion intensity (each ion peak intensity is normalized to the sum of all ion signals; positive and negative ions were analysed separately).**

Regarding organonitrate, we have found some laboratory evidence for its potential origin. AIDA (aerosol interaction and dynamics in the atmosphere) simulation chamber studies have been done with α-pinene and/or toluene OH oxidations on $NaNO_3$ seeds in the presence/absence of $NO_x$. The corresponding single particle  spectra are shown in Fig. S9. It turns out that m/z 129+ is quite weak in α-pinene system even with $NO_x$ (panel b), but it is much stronger after toluene is added (panel c). In the toluene system (refer to panel d to f), m/z 129+ is already present when $NaNO_3$ seeds are coated with toluene derived secondary organic aerosol (SOA) (panel d), likely due to the recombined fragments from toluene and nitrate. After adding extra OH, m/z 129+ is becoming stronger, likely due to the more aged toluene SOA. When NOx is added to toluene system, m/z 129+ becomes much stronger (panel e), likely due to the organonitrate formation. Hence, we can conclude that m/z 129 $C_5H_7NO_3^+$ is more related to toluene rather than α-pinene, namely m/z 129 $C_5H_7NO_3^+$ could be a signature peak for anthropogenic sources in LAAPTOF spectra.

[Figure]

Figure S11

[Figure]

**Figure S12: Time series of the estimated mass concentration of sodium salts particles measured by the LAAPTOF (a), methanesulfonic acid (MSA) fraction to total organics and sulfate measured by the AMS (b) and MSA mass concentration (c). The first (blue) and second (green) shaded areas mark the sodium chloride rich and organic rich episodes, respectively.**

The method used to estimate MSA fraction ($f_{MSA}$, signal fraction of MSA to total org and sulfate) and its mass concentration can be referred to Huang et al. (2015). In brief, W-mode $f_{MSA}$ is calculated and then used to multiply V-mode mass concentration of organics and sulfate, resulting estimated MSA mass concentration. The corresponding equations are as follows:

$$\sum I_{MSA} = \frac{I_{CH3SO2}}{9.7\%} \tag{S2}$$

$$f_{MSA} = \frac{\dfrac{\sum I_{MSA}}{RIE_{MSA}}}{\dfrac{\sum I_{Org}}{RIE_{Org}} + \dfrac{\sum I_{SO4}}{RIE_{SO4}}} \tag{S3}$$

$$C_{MSA} = f_{MSA} \times (C_{Org} + C_{SO4}) \tag{S4}$$

where $\sum I_{MSA}$ is the total signal intensity of all MSA fragments measured in W-mode, which can be estimated based on the fragmentation patterns determined from the laboratory experiments (Huang et al., 2015). It was reported that the signal intensity of marker peak at m/z 78.99 ($CH_3SO_2^+$) accounted for $(9.7 \pm 1.6)$ % of the total signal intensity of all MSA fragments. $\sum I_{Org}$ and $\sum I_{SO4}$ are total signal intensity of organics and sulfate fragments measured in W-mode, respectively. Relative ion efficiency of MSA ($RIE_{MSA}=1.3$) is the averaged value from $RIE_{Org}$ (=1.4) and $RIE_{SO4}$ (=1.2). $C_{Org}$ and $C_{SO4}$ are mass concentrations calculated from V-mode data. $C_{MSA}$ is the estimated MSA mass concentration.

[Figure]

**Figure S13**: Time series of vertical profiles for organics, ammonium, nitrate, and sulfate as calculated by COSMO-ART.

Particulate organic mass concentrations and wind field calculated by COSMO-ART from 20160827 12:00 to 20160828 01:00
30   (local time) for 10 m a.g.l.

---

## Author Response (AR2)

We gratefully thank the co-editor and referees for their helpful comments to improve the quality of our manuscript. The comments are in black. Our point-to-point replies marked by "R" are in blue. Changes to the manuscript text are in green.

**Co-editor's comments:**

**1. Line 136: The use of the PM2.5 lens on the AMS needs to be explicitly stated and perhaps cited (Peck et al., 2016) since it is not standard yet.**

R: We have cited this reference in the revised manuscript.

**2. Line 151: The software is written for Igor, it is not IGOR software.**

R: We have revised this to "standard software written for Igor"

**3. Lines 244-259: (weekday/weekend analysis) Can you please add details about weekday/weekend statistics (supplement would be fine). In particular, some indication of the observed variability in the particle class concentrations in Figs. 2 and S5 would help the reader judge the significance of the weekday/weekend claims. In particular, the Biomass burning-soil effect looks like it may be mostly driven by 1 weekend event in mid-August.**

R: We have added details about weekday/weekend statistics in the caption to Fig S5 and indicated the variability in the caption to Fig 2, as follows:

Fig S5 caption:

The data comprises 4842 time points of 10 min averaged data, including 3702 for weekdays and 1140 for weekends, including Saturday and Sunday. The total particle mass concentration for each wind direction is on average 3.8 ($\pm$ 2.3, one $\sigma$) µg m$^{-3}$ and 5.2 ($\pm$ 3.0) µg m$^{-3}$ during weekdays and the weekends, respectively. For the dominating particle classes the average mass concentrations are 1.3 ($\pm$ 1.2) µg m$^{-3}$ and 0.8 ($\pm$ 0.5) µg m$^{-3}$ for class 3, 0.2 ($\pm$ 0.5) µg m$^{-3}$ and 0.1 ($\pm$ 0.2) µg m$^{-3}$ for class 4, 1.1 ($\pm$ 1.0) µg m$^{-3}$ and 2.3 ($\pm$ 1.7) µg m$^{-3}$ for class 5, for weekdays and weekends respectively.

Fig 2 caption:

In total 4842 time points of 10 min averaged data were analysed for these plots. The total particle mass concentration for each hour of the day is on average 4.3 ($\pm$ 2.6, one $\sigma$) µg m$^{-3}$ and for each wind direction is on average 4.5 ($\pm$ 2.9) µg m$^{-3}$,. For the dominating classes the average mass concentrations are 1.3 ($\pm$ 1.1) µg m$^{-3}$ and 1.2 ($\pm$ 1.0) µg m$^{-3}$ for class 3, 0.2 ($\pm$ 0.5) µg m$^{-3}$ and 0.2 ($\pm$ 0.4) µg m$^{-3}$ for class 4, 1.4 ($\pm$ 1.0) µg m$^{-3}$ and 1.6 ($\pm$ 1.5) µg m$^{-3}$ for class 5, for (b) and (d) respectively.

**4. Line 273-275: Please clarify that the PMF analysis is for the AMS data.**

R: We have clarify this as "AMS-PMF"

**5. Lines 260-278: I find this section confusing. It seems that the conclusion at the end is that biomass burning is not a source of the biomass burning-soil particles. Is this correct? If so, would it be better to give the class a different name?**

R: The names for particle classes were only based on the LAAPTOF data analysis (different classes were directly derived from clustering for single particle data). This is discussed in detail in our previous study (Shen et al., 2019). The representative mass spectra of class 5 particles most likely feature aged biomass-burning and soil dust (Shen et al., 2019). In this study, we tried to refine the type for class 5, however, we did not achieve a consistent identification for this class even with the additional information obtained from other instruments, i.e., AMS and Aethalometer. Therefore, we would like to keep this name that is used in our previous publication (Shen et al., 2019) even though we discuss that also other particle types may contribute to this class.

**6. Lines 287-293: This acidity indicator is highly uncertain (e.g., Hennigan et al., (2015)). Given the presence of Na+, it is even more likely to be erroneous under these conditions. Please add a discussion regarding the limitations of this acidity indicator and its impacts on your conclusions.**

R: We have revised the corresponding contents in the last paragraph of section 3.1, as follows:

"For particle acidity, the molar ratio of $NH_4^+$ measured/predicted from AMS measurement, is commonly used as an indicator (Zhang et al., 2007). The predicted $NH_4^+$ value is the mass concentration required to fully neutralize sulfate, nitrate, and chloride measured by AMS. Lower ratio value represents higher acidity under certain conditions, i.e., inorganic species are dominating the particle mass and crustal elements have relatively low concentrations (Zhang et al., 2007). It should be noted that the organic acids and crustal elements such as sodium can bias the results (Hennigan et al., 2015, and the reference therein). Despite of the large uncertainty for indicating acidity, we still can use such ratio to indicate special events, e.g., sodium nitrate rich particles will provide nitrate, leading to lower values. As shown in Fig. 3, this ratio reaches lowest value during the sodium salts rich episode, while the value in the organic rich episode is at the average level. If we assume the organic acids only have minor influence on the particle acidity, the anti-correlation between acidity indicator value and organic concentrations may be related to acid-catalysed organic particle formation."

In addition, we have changed "highest acidity" to "lowest value" (for acidity indicator) in table 2

**7. Lines 294-296: NaNO3 gives a different NO+/NO2+ ratio than does ammonium nitrate. Please comment on how this could impact the interpretation of these results.**

R: We have added the comments as follows:

"It should be noted that $NaNO_3$ can also result in higher $NO^+/NO_2^+$ ratios (5 to 7, Allan et al., 2004) than $NH_4NO_3$ (2.3 in our calibration). However, the $NO^+/NO_2^+$ values here are much higher (c.f. Fig. S7) and thus we can expect a major contribution by organonitrates."

**8. Line 299: The average oxidation state does not look all that much different to me during the organic rich episode. Perhaps there is slightly less diurnal variability which results in an average state that is more oxidized? Please discuss further.**

R: Yes, the time series of oxidation state (Fig. S7) does not show much difference regarding the average, however, the statistic calculation results are -0.19 ± 0.23 for the entire campaign, -0.10 ± 0.20 for the organic rich episode,

and -0.24 ± 0.34 for sodium chloride rich episode, respectively. The latter two values are listed in Table 2 with the variability description. We have added some statement, as follows:

"It turns out that the organics were more oxidized in the organic rich episode with relatively higher $\overline{OS}_C$ value and less variability (-0.10 ± 0.20) than that for the entire campaign (-0.19 ± 0.23) (refer to Fig. S7 and Table 2)."

**9. Lines 350-360: Please see previous comment regarding NO+/NO2+ ratio. It is very difficult to judge correlation between ions from Figure S10 due to the amount of data included. Please consider showing only the time period in question.**

R: We have revised Figure S10 to show the sodium salts rich and organic rich episodes. In addition, we have added FIGAERO-HR-ToF-CIMS (stands for a filter inlet for gases and aerosols coupled to a high-resolution time-of-flight chemical ionization mass spectrometer, hereafter CIMS) results on organonitrates and modified the caption, as follows:

[Figure]

**Figure S10: Time series of sodium chloride and nitrate, and organonitrate signatures in LAAPTOF mass spectra, as well as organonitrate mass concentration derived from AMS and FIGAERO-HR-ToF-CIMS (stands for a filter inlet for gases and aerosols coupled to a high-resolution time-of-flight chemical ionization mass spectrometer, hereafter CIMS) measurement. For LAAPTOF results, Y-axis is the normalized ion intensity (each ion peak intensity is normalized to the sum of all ion signals; positive and negative ions were analysed separately). The seven classes are class 1: calcium-Soil; class 2: aged soot; class3: sodium salts; class 4: secondary inorganics-amine; class 5: biomass burning-soil; class 6: biomass burning-organosulfate; and class 7: mixed/aged-dust. Obvious sodium chloride and nitrate signatures are mainly found in class 3 labelled in blue colour. AMS and CIMS results (Pearson's correlation coefficient γ=0.52) are adapted from our companion study by Huang et al (2019). It should be noted that during the sodium chloride rich episode (first band in blue) the fraction of organonitrates is higher (Fig. S7), although the organonitrate mass concentration is lower than that during the organic rich episode (second band in green).**

**10. Lines 360-363: It seems to me premature to "confirm" that the C5H7NO3+ signal is due to organonitrates in this data given the possible issues with AMS NO/NO2.**

R: Our AIDA simulation chamber study has shown evidence that this LAAPTOF signal m/z 129 $C_5H_7NO_3+$ can arise from the recombination of inorganic nitrate and organics due to the laser ionization matrix effect and even stronger when arising from organonitrates formed under $NO_x$ rich conditions (refer to Fig. S11). In the field measurement, we have observed quite strong $C_5H_7NO_3+$ signals at the end of sodium salts rich episode, when a CIMS was also deployed. In the companion study, Huang et al. (2019) showed that AMS derived organonitrate mass concentrations were comparable with the results of molecular analysis using CIMS with Pearson's correlation coefficient γ= 0.52 (cf. Fig. S1 in Huang et al., 2019). This validates to some extent that the method for organonitrate estimation based on the AMS $NO_2+/NO+$ ratio (Farmer et al., 2010) is applicable in our study. In the revised supplement here, we have added the AMS and CIMS results for comparison. Taken together, we can conclude that m/z 129 $C_5H_7NO_3+$ was mainly resulting from organonitrates, but we cannot rule out the contribution from the recombination of inorganic nitrate and organics due to the laser ionization matrix effect.

We have revised the corresponding sentences, as follows:

"For sodium salt particles, a weak but characteristic peak at m/z 129 $C_5H_7NO_3^+$ is likely to be an organonitrate signature (Shen et al., 2019). … In the sodium chloride rich episode, in general m/z 129 $C_5H_7NO_3^+$ has a similar trend as organonitrate mass concentrations estimated by AMS results and the results of a molecular analysis using FIGAERO-HR-ToF-CIMS (stands for a filter inlet for gases and aerosols coupled to a high-resolution time-of-flight chemical ionization mass spectrometer) in a companion study by (Huang et al., 2019). … In addition, our simulation chamber study has shown that m/z 129 $C_5H_7NO_3^+$ arises from the recombination of inorganic nitrate and organics due to the laser desorption and ionization (LDI) matrix effect in single-particle MS (Murphy, 2007; Reilly et al., 2000; Reinard and Johnston, 2008; Zenobi and Knochenmuss, 1998) and even stronger when it is arising from organonitrates formed under $NO_x$ rich conditions (refer to Fig. S11). Taken together, we can conclude that the m/z 129 $C_5H_7NO_3^+$ signal is mainly resulting from organonitrates, but we still cannot rule out the LDI effect."

**11. Lines 411-420: The diurnal cycle of LV-OOA1 is difficult to see with the y-axis range given in Fig. S4. I cannot see that there are 2 peaks.**

R: We have change the y-axis range in Fig. S4 (c) and added arrows to point out the 2 peaks in the inserted figure in panel (d). Please refer to our reply to the 1st Referee comment #6.

**12. Sect 3.4: More information needs to be provided on the AMS PMF results (supplement is ok). Specifically, why were 5 factors and not 4 chosen? There is significant similarity in the three LV-OOA signals and LV-OOA3 doesn't seem to have many distinguishing features.**

R: The criteria to choose factors have been given in method section 2.2 of the main manuscript. The reasons to choose 5 factors rather than 4 are: 1) residuals (unexplained data) resulting from factor 4 is more than factor 5; 2) in spite of some similarity, distinguishable features can still be found for three LV-OOA signals, according to their time series and diurnal patterns. In particular during the organic rich episode, LV-OOA1 and LV-OOA2 show negative correlation, the LV-OOA3 signal is much weaker than the other two LV-OOA during the organic rich episode. Obvious differences can be seen in the time series shown in the figure below:

[Figure]

We have added more information in the supplement, as follows:

"The criteria for choosing factor number are given in section 2.2 of main manuscript. As shown in the Fig. S4 (a) HOA, SV-OOA, and LV-OOA have distinguishable spectra. For three LV-OOA, in spite of some similarity, distinguishable features can still be found, according to their time series (b) and diurnal patterns (c). In particular for the organic rich episode, LV-OOA1 and LV-OOA2 show anti-correlation, the LV-OOA3 signal is much weaker than the other two LV-OOA. Therefore, a five-factor solution was chosen rather than four or three-factor solution."

**13. Line 453: I don't understand what is meant by "with/without biogenic emissions" for test 1. Are the biogenic emissions included or are they not?**

R: "With/without biogenic emission test" means the comparison between the results with and without biogenic emissions. The basic model setting is with biogenic emissions, the sensitivity test is without them. Thank you for pointing out this confusing expression. We have revised it as follows:

"one without biogenic emissions (test 01)"

**14. Figure 2 caption: Can you expand on what is meant by "relatively large" and what the implications of this uncertainty is for the data interpretation.**

R: Yes. We have added more explanation in the caption as follows:

"It is worthy to note that mass quantification is one of the most challenging issues in single particle mass spectrometry (Shen et al., 2019 and the references therein). There are several assumptions mainly regarding particle shape and density in our particle mass quantification method, leading to potential uncertainties of up to ~300 %. Despite such large uncertainty, the estimated total particle mass has good correlation with the total mass measured by the AMS, covering ~85% of the measurement time (Shen et al., 2019)."

**15. Figure 7: Please consider using shading to highlight the organic peaks of interest. It will make it easier for the reader to move between the text and the figure.**

R: Thank you for your suggestion. Considering the cleanliness, we have used three green arrows to point out the peaks, as shown below:

[Figure]

**References:**

Allan, J. D., Bower, K. N., Coe, H., Boudries, H., Jayne, J. T., Canagaratna, M. R., Millet, D. B., Goldstein, A. H., Quinn, P. K., Weber, R. J., and Worsnop, D. R.: Submicron aerosol composition at Trinidad Head, California, during ITCT 2K2: Its relationship with gas phase volatile organic carbon and assessment of instrument performance, J Geophys Res-Atmos, 109, 2004.

Hennigan, C. J., Izumi, J., Sullivan, A. P., Weber, R. J., and Nenes, A.: A critical evaluation of proxy methods used to estimate the acidity of atmospheric particles, Atmospheric Chemistry and Physics, 15, 2775–2790, 2015.

Huang, W., Saathoff, H., Shen, X. L., Ramakrishna, R., Leisner, T., and Mohr, C.: Chemical characterization of highly functionalized organonitrates contributing to high night-time organic aerosol mass loadings and particle growth, Environ Sci Technol, 53, 1165–1174, 2019.

Zhang, Q., Jimenez, J. L., Worsnop, D. R., and Canagaratna, M.: A case study of urban particle acidity and its influence on secondary organic aerosol, Environ Sci Technol, 41, 3213–3219, 2007.

**Referee #1 comments:**

General comments:

The present work conducted a 6 week measurement of particle chemical composition combining LAAPToF and AMS technique at a rural site of southwest Germany. Also the regional transport model COSMO-ART was used to investigate particle sources as well as their transport pattern. The combination of measurement and model can better understand the particle sources and regional transport. The results are straightforward and useful, but some clarification and minor changes are still needed for this manuscript. I suggest a minor revision for this manuscript before publication.

Specific comments:

1. Line 18: It's a little confusing with the statement: sodium salts accounted for 36%, but organics, sulfate, nitrate, and ammonium accounted for 58%, 22%, 10%, and 9% (sum up 99%). I believe that the total mass the authors used for AMS is the sum of org and SNA. While LAAPToF measures not only non-refractory particles. It seems these results cannot be closure, especially for some readers who are not very familiar with these two instruments. Anyway, some clarification is needed to avoid misunderstanding these results.

R: Yes, the total mass used for AMS is the sum of organics and SNA, as well as chloride. In contrast, the total mass determined by LAAPTOF is the sum of 7 particle types. We have revised this sentence for clarification, as follows:

"For the entire measurement period, the total aerosol particle mass was dominated by sodium salts contributing on average $(36 \pm 27)$ % to the total single particle mass measured by the LAAPTOF. The total particulate organic compounds, sulfate, nitrate, and ammonium contributed on average $(58 \pm 12)$ %, $(22 \pm 7)$ %, $(10 \pm 1)$ %, and $(9 \pm 3)$ % to the total non-refractory particle mass measured by the AMS."

2. Line 123: Do AMS use PM2.5 lenses?

R: AMS and LAAPTOF are both are equipped with PM2.5 aerodynamic inlet lenses.

3. Line 121: It's no need for this too many references here.

R: We have removed some references here. The revised sentence is as follows:

"The LAAPTOF is a commercially available single particle mass spectrometer and has been well described in recent publications (Gemayel et al., 2016; Marsden et al., 2016; Ramisetty et al., 2018; Shen et al., 2018)."

4. Line195: Since the MS discussed long-range transport, I suggest discussing more about secondary aerosols as well.

R: The long-range transport we discuss is related to a unique episode where the sodium salts accounted for $(70 \pm 24)$ % of the total mass of the single particles measured by the LAAPTOF. In addition, the mass concentrations of organics and secondary inorganics (ammonium, sulfate, and nitrate) measured by the AMS were at the lowest level compared to the other periods during the entire measurement campaign. During the long-range transport episode secondary species mainly formed on the sodium salt particles e.g. $NaNO_3$. This is discussed regarding sub-classes and the sea salt aging process in section 3.3 "Aged sea salt in central Europe".

5. Line 188: I suggest to explain a little bit about the particle classes, not only show the names, because not all the readers have read the paper (Shen et al. 2019)

R: We have added more explanations, as follows:

"In a previous publication (Shen et al., 2019), we could identify seven major particle classes at the measurement site among ~$3.7 \times 10^5$ single particles with sizes between 200 nm and 2.5 μm $d_{va}$. In brief, Class 1 are calcium-rich and soil-dust-like particles (calcium-soil) with marker peaks at m/z 40 $Ca^+$, 56 $CaO^+$, 57 $CaOH^+$, 75 $CaCl^+$, 96 $Ca_2O^+$, and 112 $(CaO)_2^+$. This particle class shows a good correlation (Pearson's correlation coefficient $\gamma \geq 0.6$) with the laboratory derived reference spectra of soil dust particles. Class 2 are aged-soot-like particles (aged soot) with characteristic $C_n^{\pm}$ patterns and are mixed with sulfate and nitrate. Class 3 are sodium-salt-like particles (sodium salts) with marker peaks at m/z 23 $Na^+$, 81/83 $Na_2Cl^+$, 139/141 $Na_3Cl_2^+$, 165 $Na_3SO_4^+$, 35/37 $Cl^-$, 93/95 $NaCl_2^-$, 115 $Na(NO_2)_2^-$, 131 $NaNO_2NO_3^-$, and 147 $Na(NO_3)_2^-$. Class 4 are "Secondary inorganic–amine" particles with prominent ammonium, nitrate and sulfate markers, as well as amine marker peaks at m/z 58 $C_2H_5NHCH_2^+$, 59 $(CH_3)_3N^+$, 86 $(C_2H_5)_2NCH_2^+$, 88 $(C_2H_5)_2NO/C_3H_6NO_2^+$, and 118 $(C_2H_5)_2NCH_2^+$. Class 5, aged-biomass-burning and soil-dust like particles (biomass burning–soil), has aged biomass burning marker peaks at m/z 39 $K^+$, 213 $K_3SO_4^+$, 46 $NO_2^-$, 62 $NO_3^-$, 97 $HSO_4^-$, and peaks related to aromatic compounds (50 $C_4H_2^+$, 63 $C_5H_3^+$, 77 $C_6H_5^+$, 85 $C_7H^+$, 91 $C_7H_7^+$, 95 $C_7H_{11}^+$, 104 $C_8H_8^+$, 115 $C_9H_7^+$). Furthermore, class 5 shows a strong correlation with reference spectra of soil dust particles. The representative spectra of class 6, aged-biomass-burning and organosulfate-containing particles (biomass burning–organosulfate), also features aged biomass burning particles and has organosulfates marker peaks at m/z 141 $C_2H_5OSO_4^-$, 155 $C_2H_3O_2SO_4^-$, and 215 $C_5H_{11}OSO_4^-$. Class 7 particles contain all the marker peaks from the other six classes and show a good correlation with the reference spectra of mineral and soil dust particles, and are consequently labelled as "mixed/aged dust" (short for mixed/aged and dust-like particles) …"

6. Session 3.4.1 and 3.4.2: Because SOA is also an important component of fine particles, I suggest discussing more about SOA. Besides, AMS-PMF can provide OOA (SOA). This could be considered in the discussion.

R: Thank you for your suggestion. We have done a PMF analysis, added more explanation in the introduction, added one subsection in method part, added more discussion about SOA in section 3.1 and 3.4, put the corresponding results in the supplement, and modified the conclusions accordingly. The corresponding changes are listed as follows:

**Abstract**

[revised manuscript text omitted]

**Supplement**

[Figure]

**Figure S4: Five-factor-PMF solution with FPEAK=0 for organic compounds measured by AMS during the whole measurement time. (a) Mass spectra of the five components, i.e., HOA, SV-OOA, LV-OOA1, LV-OOA2, and LV-OOA3. (b) Time series of the five components mass concentrations (left Y-axis) and some related species (right Y-axis). (c) Diurnal patterns of the five components. (d) Time series of the components with the same data as panel but stacked (b). The percentages are their contributions to OA. Note that the total OA value showed here is lower than the input OA for the PMF analysis. This is because the components identification might be interfered by extraneous variability, arising from some causes such as instrumental issues. Such extraneous variability will lead to disproportionate effects on the fitting outcome. Therefore, we need to downweight the corresponding variables in a proper way or even remove them, resulting the lower output total OA value reconstructed by the components.**

The criteria for choosing factor number are given in section 2.2 of main manuscript. As shown in the Fig. S4 (a) HOA, SV-OOA, and LV-OOA have distinguishable spectra. For three LV-OOA, in spite of some similarity, distinguishable features can still be found, according to their time series (b) and diurnal patterns (c). In particular for the organic rich episode, LV-OOA1 and LV-OOA2 show anti-correlation, the LV-OOA3 signal is much weaker than the other two LV-OOA. Therefore, a five-factor solution was chosen rather than four or three-factor solution.

HOA is associated with POA, e.g., from urban emissions. In this study, the HOA (high H:C = 1.84 and low O:C = 0.12, herein) has a similar trend as m/z 57 (major contribution from $C_4H_9$) and $NO_2$ (not shown), with Pearson's correlation coefficients $\gamma = 0.5$ and 0.4, respectively. SV-OOA (O:C = 0.32) shows a very strong correlation with m/z 91 (mainly from $C_7H_7$) ($\gamma = 0.9$) and CH family ($\gamma = 0.8$). Three LV-OOA have prominent marker peaks at m/z 28 and 44, and all have high O:C ratio (> 0.7), which indicates highly oxidized or aged OA. LV-OOA1 shows strong correlations ($\gamma = 0.8$ to 1) with CH, CHO1, CHOgt1, CHN, CHO1N, and CHOgt1N. LV-OOA2 shows good correlations with CHOgt1N ($\gamma = 0.6$). LV-OOA3 shows no good correlation with any ion families, but a similar trend as sulfate ($\gamma = 0.5$).

Similar elemental ratios for HOA, SV-OOA, and LV-OOA and their relationships with tracers such as HOA with NOx and LV-OOA with secondary inorganics, have also been reported in previous AMS-PMF studies. Readers may refer to a review paper by Zhang et al. (2011).

7. Session 4 conclusion: More implication should be provided, because this work presents a method of combination of field measurement and model. Can this method be expanded to other places and time? Otherwise this work is only a very local study.

R: Yes, this combination method can be expanded to other places and time. We have added more implications in ours conclusion, as follows:

**2nd last paragraph**

"…Such field-model comparisons and corresponding sensitivity tests are useful for targeting missing sources and mechanisms in the current model and thus help its development. Such that model can be improved to become better predictive, which is useful for e.g., legislation and implementation of policies against air pollution. Consequently, model systems suitable for a reasonable environmental policy should be validated by comprehensive field studies with state-of-the-art aerosol analytics and meteorological parameters."

**Last paragraph**

"…Our results show how close the physical and chemical nature of aerosols is related to emission sources, transformation processes, and complex transport pattern. The method of combination of filed measurement and model calculation in this study can be widely used in other places in the world, in order to investigate the origin of aerosols and their interaction between transport and transformation. For example, in some pollution hot spots like Po Valley, and some other places which are subject to both biogenic and anthropogenic emissions. Besides, it is also not limited to the summer time that we studied…"

**Reply to Co-Editor and Referees**

**Referee #2 comments:**

This paper describes a six-week ground-based study of aerosol chemistry in the Rhine Valley, near Karlsruhe. Both LAAPTOF and AMS were deployed and the field data is interpreted in conjunction with COSMO-ART model. This paper is well written, and it is a valuable contribution to ACP. While the paper is somewhat closely related to the author's previous paper, I think that there is enough new material to merit publication. I have some concerns about how AMS data is used, so I recommend some major revisions before publication.

**Major comments:**

1. There are some outstanding questions about source apportionment (for example, the biomass burning class identified by LAAPTOF). I think running PMF analysis on AMS data would help answer those questions and make the analysis more complete. There is certainly enough data to do so and it is an odd choice not to include such analysis.

R: Thank you for your suggestion. We have done PMF analysis for organics data measured by the AMS, added more explanation and discussion throughout the whole manuscript, and modified the conclusion as well. Please refer to our reply to the 1$^{st}$ Referee comment #6.

In addition, we couldn't identify biomass burning organic aerosols (BBOA) for the entire measurement period, based on AMS-PMF results. We have added this point in the last second paragraph of section 3.1, as follows:

"… It should be noted that, there were no significant biomass burning organic aerosols (BBOA) identified by PMF for the entire measurement period. The marker f60 (the mass fraction of m/z 60 to the total organics) of BBOA being very close to the background fraction of 0.003 (Crippa et al., 2014; Cubison et al., 2011). In addition to biomass burning, also other potential sources for potassium and sulfate rich particles detected by LAAPTOF must be taken into consideration, such as coal combustion (Xu et al., 2018). Therefore, further analysis is required to identify the different sources contributing to this particle class."

2. To add on to the above, a unique feature of this study is that SPMS and AMS are deployed together. It would be nice to see a bit more synergy in data analysis as well. Currently, it seems like the details focus on SPMS. For example, for marine-influenced air masses, did high resolution AMS spectra show typical AMS marine markers, such as MSA?

R: Thank you for your suggestion. We have added more synergy in data analysis regarding the marine influenced air masses. In high resolution AMS spectra we have seen the peak at 78.99 $CH_3SO_2^+$, which is reported to be the marker for methanesulfonic acid (MSA) (Huang et al., 2015). Based on the method developed by Huang et al. (2015), we have also estimated the MSA fraction and mass concentrations for the entire campaign and compare them with sodium salts measured by the LAAPTOF. We have added the corresponding discussion and figure are as follows:

**Section 3.3 Aged sea salt in central Europe (last paragraph)**

"Apart from that, in high resolution AMS spectra we have observed the peak at 78.99 $CH_3SO_2^+$, which is the marker for methanesulfonic acid (MSA) from marine or coastal areas (Huang et al., 2015) although other sources cannot be excluded (Daellenbach et al., 2017). Based on the method developed by Huang et al. (2015), we have also estimated the MSA fraction (f$_{MSA}$, signal fraction of MSA to total org and sulfate) and mass concentrations

for the entire campaign (refer to Fig. S12). It turns out that during the sodium chloride rich episode, the $f_{MSA}$ is enhanced compared to the other low sodium salts time period, such as organic rich episode. This also supports that sodium salts particles measured by the LAAPTOF had sea salt origin."

**Supplement**

[Figure]

**Figure S12: Time series of the estimated mass concentration of sodium salts particles measured by the LAAPTOF (a), methanesulfonic acid (MSA) fraction to total organics and sulfate measured by the AMS (b) and MSA mass concentration (c). The first (blue) and second (green) shaded areas mark the sodium chloride rich and organic rich episodes, respectively.**

The method used to estimate MSA fraction ($f_{MSA}$, signal fraction of MSA to total org and sulfate) and its mass concentration can be referred to Huang et al. (2015). In brief, W-mode $f_{MSA}$ is calculated and then used to multiply V-mode mass concentration of organics and sulfate, resulting estimated MSA mass concentration. The corresponding equations are as follows:

$$\sum I_{MSA} = \frac{I_{CH3SO2}}{9.7\%} \tag{S2}$$

$$f_{MSA} = \frac{\dfrac{\sum I_{MSA}}{RIE_{MSA}}}{\dfrac{\sum I_{Org}}{RIE_{Org}} + \dfrac{\sum I_{SO4}}{RIE_{SO4}}} \tag{S3}$$

$$C_{MSA} = f_{MSA} \times (C_{Org} + C_{SO4}) \tag{S4}$$

where $\sum I_{MSA}$ is the total signal intensity of all MSA fragments measured in W-mode, which can be estimated based on the fragmentation patterns determined from the laboratory experiments (Huang et al., 2015). It was reported that the signal intensity of marker peak at m/z 78.99 ($CH_3SO_2^+$) accounted for $(9.7 \pm 1.6)$ % of the total signal intensity of all MSA fragments. $\sum I_{Org}$ and $\sum I_{SO4}$ are total signal intensity of organics and sulfate fragments measured in W-mode, respectively. Relative ion efficiency of MSA ($RIE_{MSA}$=1.3) is the averaged value from $RIE_{Org}$ (=1.4) and $RIE_{SO4}$ (=1.2). $C_{Org}$ and $C_{SO4}$ are mass concentrations calculated from V-mode data. $C_{MSA}$ is the estimated MSA mass concentration.

**Minor comments:**

3. Page 4, line 118: ": : :deployment of a laser ablation aerosol particles time-of-flight mass spectrometer (LAAPTOF; AeroMegt GmbH)" Pretty sure this should be "particle" instead of "particles"

R: Yes, we have revised it.

4. Page 5, line 138: Acronym TSP shows up for the first time and has not been defined.

R: We have replace Acronym TSP with the full name "total suspended particulates"

5. Page 6, line 180: "behaved anti-correlated" is an odd turn of the phrase. Would revise to "were anti-correlated"

R: We have revised it to "were anti-correlated"

6. Page 6, line 182: should this read ": : :for particles larger than 2.5 μm measured in this study: : :" instead of 2.5 nm?

R: This particle number concentration was measured by CPC3776 (TSI Inc.) which has the cut off at 2.5 nm rather than 2.5 μm. We have added this point to make it clear, as follows:

"The averaged particle number concentration of $(1.2 \pm 0.8) \times 10^4 \, cm^{-3}$ for particles larger than 2.5 nm measured by CPC3776 in this study"

7. For Figure 2 and associated discussion, are these averages over the entire campaign?

R: Yes. We have added "the data is averaged over the entire campaign" in the figure caption and discussion part (the third paragraph of section 3.1).

8. Page 8, line 246: averaged -> average

R: We have revised it.

9. Page 8, line 248: organics -> organic

R: We have revised it.

10. Figure 4 caption: extend -> extent

R: We have revised it

11. In Figure S9, it is hard to tell visually that LAAPTOF m/z 129 and AMS organonitrate are correlated. There are quite a few instances where they are not. Can you plot these against each other and give the R2?

R: We agree that it is hard to tell they are correlated. Due to complex laser desorption and ionization (LDI) mechanisms, the resulting ion fragments usually do not have good correlation or direct clear relationship with the results from quantitative instruments such as AMS. This is also the case for LAAPTOF m/z 129 and AMS organonitrate. Specific relationships of LAAPTOF ion intensities and AMS mass concentrations for non-refractory compounds were found based on specific calculation, such as the fraction of org/(org+nitrate), reported by our previous study (Shen et al., 2019). However, in this study, we aimed to confirm that the m/z 129 $C_5H_7NO_3^+$ signal is mainly due to organonitrates rather than the recombination of nitrate and organic fragments due to the LDI matrix effect in single-particle MS. First, we found that during sodium chloride rich episode, organic mass

concentration measured by the AMS was relatively low. However, at the end of this episode m/z 129 $C_5H_7NO_3+$ relative intensity was obviously strong, which was comparable with that observed in high organic episode. Furthermore, we referred to AMS results as well as CIMS results (Huang et al., 2019), both of which show high organonitrate mass concentration at the similar time period as LAAPTOF_ m/z 129 shows high relative intensity (as shown in the figured below). Therefore, we can conclude that m/z 129 $C_5H_7NO_3+$ was mainly contributed by organonitrate.

[Figure]

We have added more explanation in the last paragraph of section 3.3. Please refer to our reply to the co-editor's comment #9

**References:**

[revised manuscript text omitted]

**Figure S4: Five-factor-PMF solution with FPEAK=0 for organic compounds measured by AMS during the whole measurement time. (a) Mass spectra of the five components, i.e., HOA, SV-OOA, LV-OOA1, LV-OOA2, and LV-OOA3. Peaks are coloured with respect to different ion families. (b) Time series of the five components mass concentrations (left Y-axis) and their tracers or related ion family (right Y-axis). (c) Diurnal patterns of the five components. (d) Time series of the stacked components with the same data as panel (b), but the contribution of the five components can be visualized clearly. The percentages are their contributions to OA. The insert shows time series of SV-OOA, LV-OOA1 and LV-OOA2 during organic rich episode, and the two back arrows point out night-time and day time sub-peaks of the first LV-OOA1 peak. Note that the total OA value showed here is lower than the input OA for the PMF analysis. This is because the components identification might be interfered by extraneous variability, arising from some causes such as instrumental issues. Such extraneous variability will lead to disproportionate effects on the fitting outcome. Therefore, we need to downweight the corresponding variables in a proper way or even remove them, resulting the lower output total OA value reconstructed by the components.**

The criteria for choosing factor number are given in section 2.2 of main manuscript. As shown in the Fig. S4 (a) HOA, SV-OOA, and LV-OOA have distinguishable spectra. For three LV-OOA, in spite of some similarity, distinguishable features can still be found, according to their time series (b) and diurnal patterns (c). In particular for the organic rich episode, LV-OOA1 and LV-OOA2 show anti-correlation, the LV-OOA3 signal is much weaker than the other two LV-OOA. Therefore, a five-factor solution was chosen rather than four or three-factor solution.

HOA is associated with POA, e.g., from urban emissions. In this study, the HOA (high H:C = 1.84 and low O:C = 0.12, herein) has a similar trend as m/z 57 (major contribution from $C_4H_9$) and $NO_2$ (not shown), with Pearson's correlation coefficients $\gamma = 0.5$ and 0.4, respectively. SV-OOA (O:C = 0.32) shows a very strong correlation with m/z 91 (mainly from $C_7H_7$) ($\gamma = 0.9$) and CH family ($\gamma = 0.8$). Three LV-OOA have prominent marker peaks at m/z 28 and 44, and all have high O:C ratio (> 0.7), which indicates highly oxidized or aged OA. LV-OOA1 shows strong correlations ($\gamma = 0.8$ to 1) with CH, CHO1, CHOgt1, CHN, CHO1N, and CHOgt1N. LV-OOA2 shows good correlations with CHOgt1N ($\gamma = 0.6$). LV-OOA3 shows no good correlation with any ion families, but a similar trend as sulfate ($\gamma = 0.5$).

Similar elemental ratios for HOA, SV-OOA, and LV-OOA and their relationships with tracers such as HOA with NOx and LV-OOA with secondary inorganics, have also been reported in previous AMS-PMF studies. Readers may refer to a review paper by Zhang et al. (2011).

[Figure]

10 **Figure S5: Comparison of particle classes measured by LAAPTOF as a function of wind direction for weekdays (left) and weekends (right). The data is averaged over the entire campaign. The data comprises 4842 time points of 10 min averaged data, including 3702 for weekdays and 1140 for weekends, including Saturday and Sunday. The total particle mass concentration for each wind direction is on average 3.8 ($\pm$ 2.3, one $\sigma$) µg m$^{-3}$ and 5.2 ($\pm$ 3.0) µg m$^{-3}$ during weekdays and the weekends, respectively. For the dominating particle classes the average mass concentrations are 1.3 ($\pm$ 1.2) µg m$^{-3}$ and 0.8 ($\pm$ 0.5) µg m$^{-3}$ for class 3, 0.2 ($\pm$ 0.5) µg m$^{-3}$ and 0.1 ($\pm$ 0.2) µg m$^{-3}$**
15 **for class 4, 1.1 ($\pm$ 1.0) µg m$^{-3}$ and 2.3 ($\pm$ 1.7) µg m$^{-3}$ for class 5, for weekdays and weekends respectively.**

[Figure]

**Figure S6:** (a) Time series of aged-biomass-burning and soil-dust like particles and black carbon (black and red dots are weekend data); red dots are in organics rich episode. The shaded area marks the organics rich episode. (b) Comparison of mass concentration between biomass burning-soil and black carbon measured during the weekends. Black and red dots are the data without and with organic rich episode, respectively. (c) Diurnal pattern of particle mass concentration of biomass burning-soil (BS) and black carbon (BC) during organics rich episode, respectively.

**Figure S7:** Time series of organonitrate fraction indicator (grey dots) and oxidation state (black dots). The first (blue) and second (green) shaded areas mark the sodium chloride rich and organic rich episodes, respectively.

[Figure]

**Figure S8: Hybrid Single-Particle Lagrangian Integrated Trajectory (HYSPLIT) back-trajectory analysis for 72 hours for sodium chloride rich (a) and organics rich (b) episodes, respectively.**

**Figure S9: Spectra of the sodium salts class (Shen et al., 2019) and its sub-classes: NaCl rich (averaged from 1487 particles) and NaNO$_3$ rich (averaged from 300 particles).**

[Figure]

[Figure]

**Figure S10: Time series of sodium chloride and nitrate, and organonitrate signatures in LAAPTOF mass spectra, as well as organonitrate mass concentration derived from AMS and FIGAERO-HR-ToF-CIMS (stands for a filter inlet for gases and aerosols coupled to a high-resolution time-of-flight chemical ionization mass spectrometer, hereafter CIMS) measurement. For LAAPTOF results, Y-axis is the normalized ion intensity (each ion peak intensity is normalized to the sum of all ion signals; positive and negative ions were analysed separately). The seven classes are class 1: calcium-Soil; class 2: aged soot; class3: sodium salts; class 4: secondary inorganics-amine; class 5: biomass burning-soil; class 6: biomass burning-organosulfate; and class 7: mixed/aged-dust. Obvious sodium chloride and nitrate signatures are mainly found in class 3 labelled in blue colour. AMS and CIMS results (Pearson's correlation coefficient γ=0.52) are adapted from our companion study by Huang et al (2019). It should be noted that during the sodium chloride rich episode (first band in blue) the fraction of organonitrates is higher (Fig. S7), although the organonitrate mass concentration is lower than that during the organic rich episode (second band in green).**

The method to estimate organonitrates based on AMS measurement can be referred to Farmer et al. (2010) and the parameters used here are the same as that used by Huang et al. (2019). The result in Fig. S10 is based on our AMS measurements and the corresponding equation suggested by Farmer et al. (2010), as follows:

$$y=[(Robs-Ran)(1+Ron)]/[(Ron-Ran)(1+Robs)] \qquad\qquad (S1)$$

where y is the estimated fraction of organic nitrate in the total nitrate measured by the AMS, Robs is the observed ambient ratio of $NO_2^+/NO^+$; Ran and Ron are the $NO_2^+/NO^+$ ratio for ammonium nitrate and

organonitrate, respectively. In our study Ran=0.43 based on the measurement and Ron=0.1 based on literature data. More details can be found by Huang et al. (2019).

[Figure]

**Figure** **S11: Laboratory evidence for anthropogenic organonitrate (LAAPTOF signature m/z 129 $C_5H_7NO_3^+$). (a) to (f) are the averaged spectra from several thousand single particles, respectively. Y-axis is the normalized ion intensity (each ion peak intensity is normalized to the sum of all ion signals; positive and negative ions were analysed separately).**

Regarding organonitrate, we have found some laboratory evidence for its potential origin. AIDA (aerosol interaction and dynamics in the atmosphere) simulation chamber studies have been done with α-pinene and/or toluene OH oxidations on $NaNO_3$ seeds in the presence/absence of $NO_x$. The corresponding single particle  spectra are shown in Fig. S11. It turns out that m/z 129+ is quite weak in α-pinene system even with $NO_x$ (panel b), but it is much stronger after toluene is added (panel c). In the toluene system (refer to panel d to f), m/z 129+ is already present when $NaNO_3$ seeds are coated with toluene derived secondary organic aerosol (SOA) (panel d), likely due to the recombined fragments from toluene and nitrate. After adding extra OH, m/z 129+ is becoming stronger, likely due to the more aged toluene SOA (panel e). When NOx is added to toluene system, m/z 129+ becomes much stronger (panel f), likely due to the organonitrate formation. Hence, we can conclude that m/z 129 $C_5H_7NO_3^+$ is more related to toluene rather than α-pinene, namely m/z 129 $C_5H_7NO_3^+$ could be a signature peak for anthropogenic sources in LAAPTOF spectra.

[Figure]

Figure S11

[Figure]

**Figure S12: Time series of the estimated mass concentration of sodium salts particles measured by the LAAPTOF (a), methanesulfonic acid (MSA) fraction to total organics and sulfate measured by the AMS (b) and MSA mass concentration (c). The first (blue) and second (green) shaded areas mark the sodium chloride rich and organic rich episodes, respectively.**

The method used to estimate MSA fraction ($f_{MSA}$, signal fraction of MSA to total org and sulfate) and its mass concentration can be referred to Huang et al. (2015). In brief, W-mode $f_{MSA}$ is calculated and then used to multiply V-mode mass concentration of organics and sulfate, resulting estimated MSA mass concentration. The corresponding equations are as follows:

$$\sum I_{MSA} = \frac{I_{CH3SO2}}{9.7\%} \tag{S2}$$

$$f_{MSA} = \frac{\frac{\sum I_{MSA}}{RIE_{MSA}}}{\frac{\sum I_{Org}}{RIE_{Org}} + \frac{\sum I_{SO4}}{RIE_{SO4}}} \tag{S3}$$

$$C_{MSA} = f_{MSA} \times (C_{Org} + C_{SO4}) \tag{S4}$$

where $\sum I_{MSA}$ is the total signal intensity of all MSA fragments measured in W-mode, which can be estimated based on the fragmentation patterns determined from the laboratory experiments (Huang et al., 2015). It was reported that the signal intensity of marker peak at m/z 78.99 ($CH_3SO_2^+$) accounted for $(9.7 \pm 1.6)$ % of the total signal intensity of all MSA fragments. $\sum I_{Org}$ and $\sum I_{SO4}$ are total signal intensity of organics and sulfate fragments measured in W-mode, respectively. Relative ion efficiency of MSA ($RIE_{MSA}$=1.3) is the averaged value from $RIE_{Org}$ (=1.4) and $RIE_{SO4}$ (=1.2). $C_{Org}$ and $C_{SO4}$ are mass concentrations calculated from V-mode data. $C_{MSA}$ is the estimated MSA mass concentration.

[Figure]

**Figure S13**: Time series of vertical profiles for organics, ammonium, nitrate, and sulfate as calculated by COSMO-ART.

Particulate organic mass concentrations and wind field calculated by COSMO-ART from 20160827 12:00 to 20160828 01:00
30 (local time) for 10 m a.g.l.